# Multi-Sample Training for Neural Image Compression

**Tongda Xu**[1,2], **Yan Wang**[1,2] *, **Dailan He**[1], **Chenjian Gao**[1,3], **Han Gao**[1,4],
**Kunzan Liu**[1,5], **Hongwei Qin**[1]

[1]SenseTime Research, [2]Institute for AI Industry Research (AIR), Tsinghua University,
[3]Beihang University, [4]University of Electronic Science and Technology of China,
[5]Department of Electronic Engineering, Tsinghua University
{xutongda, wangyan}@air.tsinghua.edu.cn,
{hedailan, gaochenjian, gaohan1, liukunzan, qinhongwei}@sensetime.com

## Abstract

This paper considers the problem of lossy neural image compression (NIC). Current state-of-the-art (sota) methods adopt uniform posterior to approximate quantization noise, and single-sample pathwise estimator to approximate the gradient of evidence lower bound (ELBO). In this paper, we propose to train NIC with multiple-sample importance weighted autoencoder (IWAE) target, which is tighter than ELBO and converges to log likelihood as sample size increases. First, we identify that the uniform posterior of NIC has special properties, which affect the variance and bias of pathwise and score function estimators of the IWAE target. Moreover, we provide insights on a commonly adopted trick in NIC from gradient variance perspective. Based on those analysis, we further propose multiple-sample NIC (MS-NIC), an enhanced IWAE target for NIC. Experimental results demonstrate that it improves sota NIC methods. Our MS-NIC is plug-and-play, and can be easily extended to other neural compression tasks.

## 1 Introduction

Latent variable-based lossy neural image compression (NIC) has witnessed significant success. The majority of NIC follows the framework proposed by Ballé et al. [2017]: For encoding, the original image $x$ is transformed into $y$ by the encoder. Then $y$ is scalar-quantized into integer $\bar{y}$, estimated with an entropy model $p(\bar{y})$ and coded. For decoding, $\bar{y}$ is transformed back by the decoder to obtain reconstructed $\bar{x}$. The optimization target of NIC is R-D cost: $R + \lambda D$. $R$ denotes the bitrate of $\bar{y}$, $D$ denotes the distortion between $x$ and $\bar{x}$, and $\lambda$ denotes the hyper-parameter controlling their trade-off. During training, the quantization $\bar{y} = \lfloor y \rceil$ is relaxed with $\tilde{y} = y + \epsilon$ to simulate the quantization noise. And $\epsilon$ is fully factorized uniform noise $\epsilon \sim p(\epsilon) = \prod \mathcal{U}(-\frac{1}{2}, +\frac{1}{2})$.

Ballé et al. [2017] further recognises that such training framework is closely related to variational inference. Indeed, the above process can be formulated as a graphic model $x \leftarrow \tilde{y}$. During encoding, $x$ is transformed into variational parameter $y$ by inference model (encoder), and $\tilde{y}$ is sampled from variational posterior $q(\tilde{y}|x)$, which is a unit unifrom distribution centered in $y$. The prior likelihood $p(\tilde{y})$ is computed, and $\tilde{y}$ is transformed back by the generative model (decoder) to compute the likelihood $p(x|\tilde{y})$. Under such formulation, the prior is connected to the bitrate, the likelihood is connected to the distortion, and the posterior likelihood is connected to the bits-back bitrate (See Appendix. 2.3), which is 0 in NIC. Finally, the evidence lower bound (ELBO) is the negative $R + \lambda D$ target (Eq. 1). Denote the transform function $\tilde{y}(\epsilon; \phi) = y + \epsilon$, and sampling $\tilde{y} \sim q(\tilde{y}|x)$ is equivalent to transforming $\epsilon$ through $\tilde{y}(\epsilon; \phi)$. Then the gradient of ELBO is estimated via pathwise estimator with single-sample Monte Carlo (Eq. 2). This is the same as SGVB-1 [Kingma and Welling, 2013].

---

*Yan Wang is the corresponding author.

$$\mathcal{L} = -(R + \lambda D) = \mathbb{E}_{q(\tilde{\boldsymbol{y}}|\boldsymbol{x})}[\underbrace{\log p(\boldsymbol{x}|\tilde{\boldsymbol{y}})}_{\text{- distortion}} + \underbrace{\log p(\tilde{\boldsymbol{y}})}_{\text{- rate}} \underbrace{- \log q(\tilde{\boldsymbol{y}}|\boldsymbol{x})}_{\text{bits-back rate: 0}}] \tag{1}$$

$$\nabla_{\phi}\mathcal{L} = \mathbb{E}_{p(\boldsymbol{\epsilon})}[\nabla_{\phi}(\log \tfrac{p(\boldsymbol{x},\tilde{\boldsymbol{y}}(\boldsymbol{\epsilon};\phi))}{q(\tilde{\boldsymbol{y}}(\boldsymbol{\epsilon};\phi)|\boldsymbol{x})})] \approx \nabla_{\phi}\log \tfrac{p(\boldsymbol{x},\tilde{\boldsymbol{y}}(\boldsymbol{\epsilon};\phi))}{q(\tilde{\boldsymbol{y}}(\boldsymbol{\epsilon};\phi)|\boldsymbol{x})} \tag{2}$$

Ballé et al. [2018] further extends this framework into a two-level hierarchical structure, with graphic model $\boldsymbol{x} \leftarrow \tilde{\boldsymbol{y}} \leftarrow \tilde{\boldsymbol{z}}$. The variational posterior is fully factorized uniform distribution $\mathcal{U}(\boldsymbol{y} - \frac{1}{2}, \boldsymbol{y} + \frac{1}{2})\mathcal{U}(\boldsymbol{z} - \frac{1}{2}, \boldsymbol{z} + \frac{1}{2})$ To simulate the quantization noise. And $\boldsymbol{y}, \boldsymbol{z}$ denote outputs of their inference networks.

$$\mathcal{L} = \mathbb{E}_{q(\tilde{\boldsymbol{y}},\tilde{\boldsymbol{z}}|\boldsymbol{x})}[\underbrace{\log p(\boldsymbol{x}|\tilde{\boldsymbol{y}})}_{\text{- distortion}} + \underbrace{\log p(\tilde{\boldsymbol{y}}|\tilde{\boldsymbol{z}}) + \log p(\tilde{\boldsymbol{z}})}_{\text{- rate}} \underbrace{- \log q(\tilde{\boldsymbol{y}}|\boldsymbol{x}) - \log q(\tilde{\boldsymbol{z}}|\tilde{\boldsymbol{y}})}_{\text{bits-back rate: 0}}] \tag{3}$$

The majority of later NIC follows this hierarchical latent framework [Minnen et al., 2018, Cheng et al., 2020]. Some focus on more expressive network architectures [Zhu et al., 2021, Xie et al., 2021], some stress better context models [Minnen and Singh, 2020, He et al., 2021, Guo et al., 2021a], and some emphasize semi-amortization inference [Yang et al., 2020]. However, there is little research on multiple-sample methods, or other techniques for a tighter ELBO.

On the other hand, IWAE [Burda et al., 2016] has been successful in density estimation. Specifically, IWAE considers a multiple-sample lowerbound $\mathcal{L}_k$ (Eq. 4), which is tighter than its single-sample counterpart. The benefit of such bound is that the implicit distribution defined by IWAE approaches true posterior as $k$ increases [Cremer et al., 2017]. This suggests that its variational posterior is less likely to collapse to a single mode of true posterior, and the learned representation is richer. The gradient of $\mathcal{L}_k$ is computed via pathwise estimator. Denote the exponential ELBO sample as $w_i$, its reparameterization as $w(\boldsymbol{\epsilon}_i; \phi)$, and its weight $\tilde{w}_i = \frac{w_i}{\sum w_j}$. Then $\nabla_{\phi}\mathcal{L}_k$ has the form of importance weighted sum (Eq. 5).

$$\mathcal{L}_k = \mathbb{E}_{q(\tilde{\boldsymbol{y}}_{1:k}|\boldsymbol{x})}[\log \tfrac{1}{k}\sum_i^k \underbrace{\tfrac{p(\boldsymbol{x},\tilde{\boldsymbol{y}}_i)}{q(\tilde{\boldsymbol{y}}_i|\boldsymbol{x})}}_{w_i}] = \mathbb{E}_{p(\boldsymbol{\epsilon}_{1:k})}[\log \tfrac{1}{k}\sum_i^k \underbrace{\tfrac{p(\boldsymbol{x},\tilde{\boldsymbol{y}}(\boldsymbol{\epsilon}_i;\phi))}{q(\tilde{\boldsymbol{y}}(\boldsymbol{\epsilon}_i;\phi)|\boldsymbol{x})}}_{w(\boldsymbol{\epsilon}_i;\phi)}] \tag{4}$$

$$\nabla_{\phi}\mathcal{L}_k = \mathbb{E}_{p(\boldsymbol{\epsilon}_{1:k})}[\sum_i^k \tilde{w}_i \nabla_{\phi} \log w(\boldsymbol{\epsilon}_i;\phi)] \approx \sum_i^k \tilde{w}_i \nabla_{\phi} \log w(\boldsymbol{\epsilon}_i;\phi) \tag{5}$$

In this paper, we consider the problem of training NIC with multiple-sample IWAE target (Eq. 4), which allows us to learn a richer latent space. First, we recognise that NIC's factorized uniform variational posterior has impacts on variance and bias properties of gradient estimators. Specifically, we find NIC's pathwise gradient estimator equivalent to an improved STL estimator [Roeder et al., 2017], which is unbiased even for the IWAE target. However, NIC's IWAE-DReG estimator [Tucker et al., 2018] has extra bias, which causes performance decay. Moreover, we provide insights on a commonly adopted but little explained trick of training NIC from gradient variance perspective. Based on those analysis and observations, we further propose MS-NIC, a novel improvement of multiple-sample IWAE target for NIC. Experimental results show that it improves sota NIC methods [Ballé et al., 2018, Cheng et al., 2020] and learns richer latent representation. Our method is plug-and-play, and can be extended into neural video compression.

To wrap up, our contributions are as follows:

- We provide insights on the impact of the uniform variational posterior upon gradient estimators, bits-back coding and a commonly adopted but little discussed trick of NIC training from gradient variance perspective.

- We propose multiple-sample neural image compression (MS-NIC). It is a novel enhancement of hierarchical IWAE [Burda et al., 2016] for neural image compression. To the best of our knowledge, we are the first to consider a tighter ELBO for training neural image compression.

- We demonstrate the efficiency of MS-NIC through experimental results on sota NIC methods. Our method is plug-and-play for neural image compression and can be easily applied to neural video compression.

## 2 Gradient Estimation for Neural Image Compression

The common NIC framework (Eq. 1, Eq 3) adopts fully factorized uniform distribution $q(\tilde{\boldsymbol{y}}, \tilde{\boldsymbol{z}}|\boldsymbol{x}) = \prod \mathcal{U}(y^i - \frac{1}{2}, y^i + \frac{1}{2}) \prod \mathcal{U}(z^j - \frac{1}{2}, z^j + \frac{1}{2})$ to simulate the quantization noise. Such formulation has the following special properties:

- Property I: $q(\tilde{\boldsymbol{z}}|\tilde{\boldsymbol{y}})$ and $q(\tilde{\boldsymbol{y}}|\boldsymbol{x})$'s support depends on the parameter.
- Property II: $\log q(\tilde{\boldsymbol{z}}|\tilde{\boldsymbol{y}}) = \log q(\tilde{\boldsymbol{y}}|\boldsymbol{x}) = 0$ on their support.

The impacts of these two properties are frequently neglected in previous works, which does not influence the results for single-sample pathwise gradient estimators (a.k.a. reparameterization trick in Kingma and Welling [2013]). In this section, we discuss the impacts of these two properties upon the variance and biasness of gradient estimators. Our analysis is based on single level latent (Eq. 1) instead of hierarchical latent (Eq. 3) to simplify notations.

### 2.1 Impact on Pathwise Gradient Estimators

First, let's consider the single-sample case. We can expand the pathwise gradient of ELBO in Eq. 2 into Eq. 6. As indicated in the equation, $\phi$ contributes to $\mathcal{L}$ in two ways. The first way is through the reparametrized $\tilde{\boldsymbol{y}}(\epsilon; \phi)$ (pathwise term), and the other way is through the parameter of $\log q(\tilde{\boldsymbol{y}}|\boldsymbol{x})$ (parameter score term). Generally, the parameter score term has higher variance than the pathwise term. The STL [Roeder et al., 2017] reduces the gradient by dropping the score. It is unbiased since the dropped term's expectation $\mathbb{E}_{q(\tilde{\boldsymbol{y}}|\boldsymbol{x})}[\nabla_\phi \log q_\phi(\tilde{\boldsymbol{y}}|\boldsymbol{x})]$ is 0.

$$\nabla_\phi \mathcal{L} = \mathbb{E}_{p(\epsilon)}[\underbrace{\nabla_{\tilde{\boldsymbol{y}}}(\log \frac{p(\boldsymbol{x}|\tilde{\boldsymbol{y}})p(\tilde{\boldsymbol{y}})}{q(\tilde{\boldsymbol{y}}|\boldsymbol{x})})\nabla_\phi \tilde{\boldsymbol{y}}(\epsilon; \phi)}_{\text{pathwise term}} - \underbrace{\nabla_\phi \log q_\phi(\tilde{\boldsymbol{y}}|\boldsymbol{x})}_{\text{parameter score term}}] \tag{6}$$

Now let's consider the STL estimator of multiple-sample IWAE bound (Eq. 4). As shown in Tucker et al. [2018], the STL estimation of IWAE bound gradient is biased. To reveal the reason, consider expanding the gradient Eq. 5 into partial derivatives as we expand Eq. 2 into Eq. 6. Unlike single-sample case, the dropped parameter score term $\mathbb{E}_{p(\epsilon_{1:k})}[\sum \tilde{w}_i(-\nabla_\phi \log q_\phi(\tilde{\boldsymbol{y}}|\boldsymbol{x}))]$ is no longer 0 due to the importance weight $\tilde{w}_i$. This means that STL loses its unbiasness in general IWAE cases.

Regarding NIC, however, the direct pathwise gradient for IWAE bound is automatically an unbiased STL estimator. Property II means that variational posterior has constant entropy, which further means that the parameter score gradient is 0. So, NIC's pathwise gradient of IWAE bound is equvailent to an extended, unbiased STL estimator.

### 2.2 Impact on Score Function Gradient Estimators

In previous section, we show the bless of NIC's special properties on pathwise gradient estimators. In this section, we show their curse on score function gradient estimators. Sepcifically, Property I implies that $q(\tilde{\boldsymbol{z}}|\tilde{\boldsymbol{y}})$ and $q(\tilde{\boldsymbol{y}}|\tilde{\boldsymbol{x}})$ are not absolute continuous, and hence the score function gradient estimators of those distributions are biased.

For example, consider a univariate random variable $x \sim p_\theta(x) = \mathcal{U}(\theta - \frac{1}{2}, \theta + \frac{1}{2})$. Our task is to estimate the gradient of a differentiable function $f(x)$. And consider the $\theta$-independent random variable $\epsilon \sim p(\epsilon) = \mathcal{U}(-\frac{1}{2}, +\frac{1}{2})$, the transform $x(\epsilon; \theta) = \theta + \epsilon$. Under such conditions, the Monte Carlo estimated pathwise gradient and score function gradient are:

$$\text{pathwise gradient: } \nabla_\theta \mathbb{E}_{p_\theta(x)}[f(x)] = \nabla_\theta \mathbb{E}_{p(\epsilon)}[f(x(\epsilon; \theta))] \approx \frac{1}{N}\sum_i^N \nabla_\theta f(\theta + \epsilon_i) \tag{7}$$

$$\text{score function gradient: } \nabla_\theta \mathbb{E}_{p_\theta(x)}[f(x)] = \mathbb{E}_{p_\theta(x)}[\nabla_\theta \log p_\theta(x)f(x)] = 0 \tag{8}$$

Eq. 7 does not equal to Eq. 8, and Eq.8 is wrong. The score function gradient is only unbiased when the distribution satisfies the absolute continuity condition of [Mohamed et al., 2020]. This reflects that under the formulation of NIC, the equivalence between the score function gradient (a.k.a. REINFORCE [Williams, 1992]) and pathwise gradient (a.k.a reparameterization trick in [Kingma and Welling, 2013]) no longer holds.

Table 1: Effect of DReG gradient estimator in NIC.

|                                  | Sample Size | bpp    | MSE   | PSNR (db) | R-D cost |
|----------------------------------|-------------|--------|-------|-----------|----------|
| *Single-sample*                  |             |        |       |           |          |
| Baseline [Ballé et al., 2018]    | -           | 0.5273 | 32.61 | 33.28     | 1.017    |
| *Multiple-sample*                |             |        |       |           |          |
| MS-NIC-MIX(pathwise gradient)    | 5           | 0.5259 | 31.84 | 33.38     | 1.003    |
| MS-NIC-MIX(DReG gradient)        | 5           | 0.5316 | 35.09 | 32.90     | 1.058    |

Such equivalence is the cornerstone of many gradient estimators, and IWAE-DReG [Tucker et al., 2018] is one of them. IWAE-DReG is a popular gradient estimator for IWAE target (Eq. 4) as it resolves the vanish of inference network gradient SNR (signal to noise ratio). However, the correctness of IWAE-DReG depends on the equivalence between the score function gradient and pathwise gradient, which does not hold for NIC. Specifically, IWAE-DReG expand the total derivative of IWAE target as Eq. 9 and perform another round of reparameterization on the score function term as Eq. 10 to further reduce the gradient variance. However, Eq. 10 requires the equivalence of pathwise gradient and score function gradient.

$$\nabla_\phi \mathbb{E}_{q_\phi(\tilde{\boldsymbol{y}}_{1:k}|\boldsymbol{x})}[\log \frac{1}{k}\sum_{i=1}^{k} w_i] = \mathbb{E}_{p(\boldsymbol{\epsilon}_{1:k})}[\sum_{i=1}^{k} \underbrace{\frac{w_i}{\sum_{j=1}^{k} w_j}(-\frac{\partial \log q_\phi(\tilde{\boldsymbol{y}}_i|\boldsymbol{x})}{\partial \phi}}_{\text{score function term}} + \frac{\partial \log w(\boldsymbol{\epsilon}_i;\phi)}{\partial \tilde{\boldsymbol{y}}_i}\frac{\partial \tilde{\boldsymbol{y}}(\boldsymbol{\epsilon}_i;\phi)}{\partial \phi})]$$

$$(9)$$

$$\mathbb{E}_{q(\tilde{\boldsymbol{y}}_i|\boldsymbol{x})}[\frac{w_i}{\sum_{j=1}^{k} w_j}\frac{\partial \log q_\phi(\tilde{\boldsymbol{y}}_i|\boldsymbol{x})}{\partial \phi}] = \mathbb{E}_{p(\boldsymbol{\epsilon}_i)}[\frac{\partial}{\partial \tilde{\boldsymbol{y}}_i}(\frac{w_i}{\sum_{j=1}^{k} w_j})\frac{\partial \tilde{\boldsymbol{y}}(\boldsymbol{\epsilon}_i;\phi)}{\partial \phi_i}] \qquad (10)$$

As we show empirically in Tab. 1, blindly adopting IWAE-DReG estimator for multiple-sample NIC brings evident performance decay. Other than IWAE-DReG, many other graident estimators such as NVIL [Mnih and Gregor, 2014], VIMCO [Mnih and Rezende, 2016] and GDReG [Bauer and Mnih, 2021] do not apply to NIC. They either bring some extra bias or are totally wrong.

## 2.3 Impact on Bits-Back Coding

It is well known that the ELBO $\mathcal{L}$ is the minus overall bitrate for bits-back coding in compression [Hinton and Van Camp, 1993, Hinton et al., 1995, Chen et al., 2017], and the entropy of variational posterior is exactly the bits-back rate itself. For this reason, earlier works [Townsend et al., 2018, Yang et al., 2020] point out that [Ballé et al., 2018, Minnen et al., 2018] waste bits for not using bits-back coding on $\boldsymbol{z}$. However, during training the differential entropy [Cover, 1999] $\mathbb{E}_{q(\tilde{\boldsymbol{z}}|\tilde{\boldsymbol{y}})}[\log q(\tilde{\boldsymbol{z}}|\tilde{\boldsymbol{y}})]$ is constant. And this means that this term does not have impact on the optimization procedure. And due to the deterministic inference, the $\log q(\tilde{\boldsymbol{z}}|\bar{\boldsymbol{y}})$ is 0, which means that the bitrate saved by bits-back coding is 0. In this sense, [Ballé et al., 2018, Minnen et al., 2018] is also optimal in bits-back coding perspective, although no actual bits-back coding is performed. In fact, there is no space for bits-back coding so long as encoder is deterministic. Since we can view deterministic encoder as a posterior distribution with mass 1 on a single point. And then the posterior's entropy is always 0.

## 2.4 The *direct-y* Trick in Training NIC

In NIC, we feed deterministic parameter $\boldsymbol{y}$ into z inference model instead of noisy samples $\tilde{\boldsymbol{y}}$. This implies that $\tilde{\boldsymbol{z}}$ is sampled from $q(\tilde{\boldsymbol{z}}|\boldsymbol{y})$ instead of $q(\tilde{\boldsymbol{z}}|\tilde{\boldsymbol{y}})$. This trick is initially adopted in Ballé et al. [2018] and followed by most of the subsequent works. However, it is little discussed. In this

paper, we refer it to *direct-y* trick. Yang et al. [2020] observes that feeding $\tilde{\boldsymbol{y}}$ instead of $\boldsymbol{y}$ causes severe performance decay. We confirm this result in Tab. 2. Thus, *direct-y* trick is essential to train hierarchical NIC.

Table 2: Effects of *direct-y* on R-D performance. 2-level VAE is equivalent to Ballé et al. [2018] without *direct-y*.

|  | bpp | MSE | PSNR | R-D cost |
|---|---|---|---|---|
| 2-level VAE | 0.9968 | 33.08 | 33.22 | 1.493 |
| [Ballé et al., 2018] | 0.5273 | 32.61 | 33.28 | 1.017 |

Table 3: Effects of *direct-y* on gradient SNR of different parts of the model. 2-level VAE is equivalent to Ballé et al. [2018] without *direct-y*. "early" is $5 \times 10^4$ iterations, "mid" is $5 \times 10^5$ iterations and "late" is $1 \times 10^6$ iterations. "infer" is the abbreviation for "inference model", and "gen" is the abbreviation for "generative model".

| Iteration | Method | gradient SNR of # | | | | |
|---|---|---|---|---|---|---|
|  |  | y infer | y gen | z infer | z gen | z prior |
| early | 2-level VAE | 2.287 | 0.5343 | 0.3419 | 0.4099 | 0.9991 |
|  | Ballé et al. [2018] | 2.174 | 0.5179 | 0.5341 | 0.3813 | 1.069 |
| mid | 2-level VAE | 1.350 | 0.4793 | 0.2414 | 0.3583 | 0.8861 |
|  | Ballé et al. [2018] | 1.334 | 0.4813 | 0.4879 | 0.3761 | 0.9693 |
| late | 2-level VAE | 1.217 | 0.4746 | 0.2863 | 0.3439 | 0.8691 |
|  | Ballé et al. [2018] | 1.206 | 0.4763 | 0.5506 | 0.3707 | 0.9339 |

One explanation is to view $q(\tilde{\boldsymbol{z}}|\boldsymbol{y})$ as $q(\tilde{\boldsymbol{z}}|\boldsymbol{x})$, and $q(\tilde{\boldsymbol{y}}, \tilde{\boldsymbol{z}}|\tilde{\boldsymbol{x}})$ factorized as $q(\tilde{\boldsymbol{y}}|\boldsymbol{x})q(\tilde{\boldsymbol{z}}|\boldsymbol{x})$ (See Fig.1 (a)-(c)). A similar trick of feeding mean parameter can be traced back to the Helmholtz machine [Dayan et al., 1995]. However, this provides a rationale why *direct-y* is fine to be adopted but does not explain why samping $\tilde{\boldsymbol{z}}$ from $q(\tilde{\boldsymbol{z}}|\tilde{\boldsymbol{y}})$ fails. We provide an alternative explanation from the gradient variance perspective. Specifically, $q(\tilde{\boldsymbol{z}}|\tilde{\boldsymbol{y}})$ has two stochastic arguments that could cause high variance in the gradient of z inference model, and make its convergence difficult. To verify this, we follow Rainforth et al. [2018] to compare the gradient SNR, which is the absolute value of the empirical mean divided by standard deviation. We trace the gradient SNR during different training stages as model converges (See Sec. 5.1 for detailed setups).

As demonstrated in Tab. 3, the gradient SNR of z inference model of standard 2-level VAE (without *direct y*) is indeed significantly lower than Ballé et al. [2018] (with *direct y*) during all 3 stage of training. This result reveals that the z inference model is more difficult to train without *direct-y*. And such difficulty could be the source of the failure of NIC without *direct-y* trick.

## 3 Multiple-sample Neural Image Compression

In this section, we consider the multiple-sample approach based on the 2-level hierarchical framework by Ballé et al. [2018], which is the de facto NIC architecture adopted by many sota methods. To simplify notations, $\log q(\tilde{\boldsymbol{z}}|\tilde{\boldsymbol{y}})$ and $\log q(\tilde{\boldsymbol{y}}|\boldsymbol{x})$ in ELBO are omitted as they are 0. First, let's consider directly applying 2-level IWAE to NIC without *direct-y* trick (See Fig. 1 (d)). Regarding a $k$ sample IWAE, we first compute parameter $\boldsymbol{y}$ of $q(\tilde{\boldsymbol{y}}|\boldsymbol{x})$ and sample $\tilde{\boldsymbol{y}}_{1:k}$ from it. Then, we compute parameter $\boldsymbol{z}_{1:k}$ of $q(\tilde{\boldsymbol{z}}_{1:k}|\tilde{\boldsymbol{y}}_{1:k})$ and samples $\tilde{\boldsymbol{z}}_{1:k}$ from it. Afterward, $\tilde{\boldsymbol{y}}_{1:k}$ and $\tilde{\boldsymbol{z}}_{1:k}$ are fed into the generative model and compute $w_{1:k}$. Finally, we follow Eq 5 to compute the gradient and update parameters. In fact, this is the standard 2-level IWAE in the original IWAE paper.

However, the vanilla 2-level IWAE becomes a problem for NIC with *direct-y* trick. Concerning a $k$ sample IWAE, we sample $\tilde{\boldsymbol{y}}_{1:k}$ from $q(\tilde{\boldsymbol{y}}|x)$. Due to the *direct-y* trick, we feed $\boldsymbol{y}$ instead of $\tilde{\boldsymbol{y}}_{1:k}$ into z inference network, and our $q(\tilde{\boldsymbol{z}}|\boldsymbol{y})$ has only one parameter $\boldsymbol{z}$ other than $k$ parameter $\boldsymbol{z}_{1:k}$. If we follow the 2-level IWAE approach, only one sample $\tilde{\boldsymbol{z}}$ is obtained, and $w_{1:k}$ can not be computed.

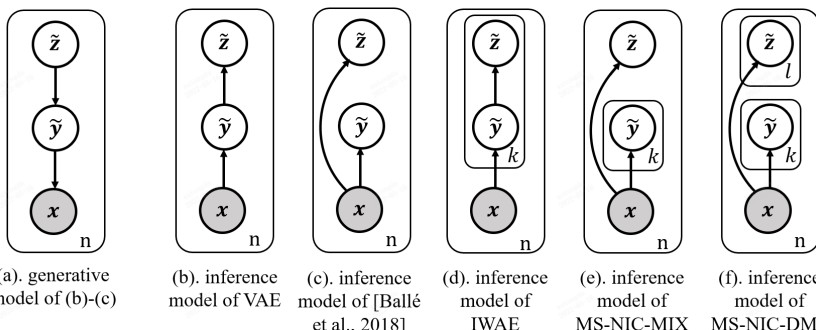

| (a). generative model of (b)-(c) | (b). inference model of VAE | (c). inference model of [Ballé et al., 2018] | (d). inference model of IWAE | (e). inference model MS-NIC-MIX | (f). inference model of MS-NIC-DMS |

Figure 1: The plate notation of different NIC methods. $\boldsymbol{x}$ is the observed image, $\tilde{\boldsymbol{y}}$ and $\tilde{\boldsymbol{z}}$ are latent. The inference models show how we sample from variational posterior duing training. $n$ is the number of data points in dataset, $k, l$ is the sample size of multiple-sample approaches. The generative model of (b), (c) is (a). The generative model of (d)-(f) is shown in Appendix. A.1. For clarity, we omit the parameters.

One method is to limit the multiple-sample part to $\tilde{\boldsymbol{y}}$ related term only and optimize other parts via single-sample SGVB-1, which produces our MS-NIC-MIX (See Fig 1 (e)). Another method is to sample another $l$ samples of $\tilde{\boldsymbol{z}}_j$ from $q(\tilde{\boldsymbol{z}}|\boldsymbol{y})$ and nest it with MS-NIC-MIX, which generates our MS-NIC-DMS (See Fig 1 (f)).

### 3.1 MS-NIC-MIX: Multiple-sample NIC with Mixture

One way to optimize multiple-sample IWAE target of NIC with *direct-y* trick is to sample $\tilde{\boldsymbol{y}}$ $k$ times to obtain $\tilde{\boldsymbol{y}}_{1:k}$ and $\tilde{\boldsymbol{z}}$ only 1 time. Then we perform $k$ sample log mean of $p(\boldsymbol{x}|\tilde{\boldsymbol{y}}_i)p(\tilde{\boldsymbol{y}}_i|\tilde{\boldsymbol{z}})$ to obtain a multiple-sample estimated $\log p(\boldsymbol{x}|\tilde{\boldsymbol{z}})$, add it with single-sample $\log p(\tilde{\boldsymbol{z}})$. This brings a $\mathcal{L}_k^{MIX}$ with the form of a mixture of 1-level VAE and 1-level IWAE ELBO:

$$\mathcal{L}_k^{MIX} = \mathbb{E}_{q_\phi(\tilde{\boldsymbol{z}}|\boldsymbol{x})}[\mathbb{E}_{q_\phi(\tilde{\boldsymbol{y}}_{1:k}|\boldsymbol{x})}[\log \frac{1}{k}\sum_i^k p(\boldsymbol{x}|\tilde{\boldsymbol{y}}_i)p(\tilde{\boldsymbol{y}}_i|\tilde{\boldsymbol{z}})|\tilde{\boldsymbol{z}}] + \log p(\tilde{\boldsymbol{z}})] \tag{11}$$

Moreover, $\mathcal{L}_k^{MIX}$ is a reasonably preferable target over ELBO as it satisfies the following properties (See Appendix. A.2 for proof):

1. $\log p(x) \geq \mathcal{L}_k^{MIX}$
2. $\mathcal{L}_k^{MIX} \geq \mathcal{L}_m^{MIX}$ for $k \geq m$

Although $\mathcal{L}_k^{MIX}$ does not converge to true $\log p(\boldsymbol{x})$ as $k$ grows, it is still a lower bound of $\log p(\boldsymbol{x})$ and tighter than ELBO (as $\mathcal{L}_1^{MIX}$ = ELBO). Its gradient can be computed via pathwise estimator. Denote the per-sample integrand $p(\boldsymbol{x}|\tilde{\boldsymbol{y}}_i)p(\tilde{\boldsymbol{y}_i}|\tilde{\boldsymbol{z}})$ as $w_i^{MIX}$, and its relative weight as $\tilde{w}_i^{MIX}$, then the gradient $\nabla_\phi \mathcal{L}_k^{MIX}$ can be estimated as Eq. 13.

$$\begin{aligned}\mathcal{L}_k^{MIX} &= \mathbb{E}_{p(\boldsymbol{\epsilon}_{1:k}^y, \boldsymbol{\epsilon}^z)}[\log \frac{1}{k}\sum_i^k p(\boldsymbol{x}|\tilde{\boldsymbol{y}}(\boldsymbol{\epsilon}_i^y; \phi))p(\tilde{\boldsymbol{y}}(\boldsymbol{\epsilon}_i^y; \phi)|\tilde{\boldsymbol{z}}(\boldsymbol{\epsilon}^z; \phi)) + \log p(\tilde{\boldsymbol{z}}(\boldsymbol{\epsilon}^z; \phi))] \\ &\approx \log \frac{1}{k}\sum_i^k \underbrace{p(\boldsymbol{x}|\tilde{\boldsymbol{y}}(\boldsymbol{\epsilon}_i^y; \phi))p(\tilde{\boldsymbol{y}}(\boldsymbol{\epsilon}_i^y; \phi)|\tilde{\boldsymbol{z}}(\boldsymbol{\epsilon}^z; \phi))}_{w^{MIX}(\boldsymbol{\epsilon}_{1:k}^y, \boldsymbol{\epsilon}^z; \phi)} + \log p(\tilde{\boldsymbol{z}}(\boldsymbol{\epsilon}^z; \phi))\end{aligned} \tag{12}$$

$$\nabla_\phi \mathcal{L}_k^{MIX} \approx \sum_i^k \tilde{w}_i^{MIX} \nabla_\phi \log w^{MIX}(\boldsymbol{\epsilon}_{1:k}^y, \boldsymbol{\epsilon}^z; \phi) + \nabla_\phi \log p(\tilde{\boldsymbol{z}}(\boldsymbol{\epsilon}^z; \phi)) \tag{13}$$

Another way to understand MS-NIC-MIX is to view the y inference/generative model as a single level IWAE, and the z inference/generative model as a large prior of $\tilde{\boldsymbol{y}}$ which is optimized via SGVB-1.

This perspective is often taken by works in NIC context model [Minnen et al., 2018, He et al., 2021], as the context model of NIC is often limited to $\tilde{y}$.

## 3.2   MS-NIC-DMS: Multiple-sample NIC with Double Multiple Sampling

An intuitive improvement over MS-NIC-MIX is to add another round of multiple-sample over $\tilde{z}$. Specifically, we sample $\tilde{z}$ $l$ times, nest it with $\mathcal{L}_k^{MIX}$ to obtain $\mathcal{L}_{k,l}^{DMS}$:

$$\mathcal{L}_{k,l}^{DMS} = \mathbb{E}_{q_\phi(\tilde{\boldsymbol{z}}_{1:l}|\boldsymbol{x})}[\log \tfrac{1}{l}\sum_j^l \exp\left(\mathbb{E}_{q_\phi(\tilde{\boldsymbol{y}}_{1:k}|\boldsymbol{x})}[\log \tfrac{1}{k}\sum_i^k p(\boldsymbol{x}|\tilde{\boldsymbol{y}}_i)p(\tilde{\boldsymbol{y}}_i|\tilde{\boldsymbol{z}}_j)|\tilde{\boldsymbol{z}}_j])p(\tilde{\boldsymbol{z}}_j)] \tag{14}$$

And we name it MS-NIC-DMS as it adopts multiple sampling twice. Moreover, $\mathcal{L}_{k,l}^{DMS}$ is a reasonably better target for optimizaion over ELBO and $\mathcal{L}_k^{MIX}$, as it satisfies the following properties (See proof in Appendix. A.2):

1. $\log p(\boldsymbol{x}) \geq \mathcal{L}_{k,l}^{DMS}$

2. $\mathcal{L}_{k,l}^{DMS} \geq \mathcal{L}_{m,n}^{DMS}$ for $k \geq m, l \geq n$

3. $\mathcal{L}_{k,l}^{DMS} \geq \mathcal{L}_k^{MIX}$

4. $\mathcal{L}_{k,l}^{DMS} \to \log p(\boldsymbol{x})$ as $k, l \to \infty$, under the assumption that $\log \frac{p(\boldsymbol{x}|\tilde{\boldsymbol{y}}_i)p(\tilde{\boldsymbol{y}}_i|\tilde{\boldsymbol{z}}_j)}{q(\tilde{\boldsymbol{y}}_i|\boldsymbol{x})}$ and $\log \frac{p(\boldsymbol{x}|\tilde{\boldsymbol{z}}_j)p(\tilde{\boldsymbol{z}}_j)}{q(\tilde{\boldsymbol{z}}_j|\boldsymbol{x})}$ are bounded.

In other words, the target $\mathcal{L}_{k,l}^{DMS}$ is a lowerbound of $\log p(\boldsymbol{x})$, converging to $\log p(\boldsymbol{x})$ as $k, l \to \infty$, tighter than $\mathcal{L}_k^{MIX}$ and tighter than ELBO (as $\mathcal{L}_{1,1}^{DMS} =$ ELBO). However, its Monte Carlo estimation is biased due to the nested transformation and expectation. Empirically, we find that directly adopting biased pathwise estimator works fine. And its gradient can be estimated by pathwise estimator similar to original IWAE target (See Eq. 5).

$$\mathcal{L}_{k,l}^{DMS} = \mathbb{E}_{q(\boldsymbol{\epsilon}_{1:l}^z)}[\log \tfrac{1}{l}\sum_j^l \exp(\mathbb{E}_{q(\boldsymbol{\epsilon}_{k:l}^y)}[\log \tfrac{1}{k}\sum_i^k p(\boldsymbol{x}|\tilde{\boldsymbol{y}}(\boldsymbol{\epsilon}_i^y;\phi))p(\tilde{\boldsymbol{y}}(\boldsymbol{\epsilon}_i^y;\phi)|\tilde{\boldsymbol{z}}(\boldsymbol{\epsilon}_j^z;\phi))])p(\tilde{\boldsymbol{z}}(\boldsymbol{\epsilon}_j^z;\phi))]$$

$$\approx \log \tfrac{1}{l}\sum_j^l \tfrac{1}{k}\sum_i^k \underbrace{p(\boldsymbol{x}|\tilde{\boldsymbol{y}}(\boldsymbol{\epsilon}_i^y;\phi))p(\tilde{\boldsymbol{y}}(\boldsymbol{\epsilon}_i^y;\phi)|\tilde{\boldsymbol{z}}(\boldsymbol{\epsilon}_j^z;\phi))p(\tilde{\boldsymbol{z}}(\boldsymbol{\epsilon}_j^z;\phi))}_{w^{DMS}(\boldsymbol{\epsilon}_{1:k}^y, \boldsymbol{\epsilon}_{1:l}^z;\phi)}$$

$$\tag{15}$$

Another interpretation of MS-NIC-DMS is to view it as a multiple level IWAE with repeated local samples. The $\mathcal{L}_{k,l}^{DMS}$ Monte Carlo pathwise estimator has the form of IWAE with $k \times l$ samples. However, there are multiple repeated samples that contain the same $\tilde{\boldsymbol{y}}_i$ and $\tilde{\boldsymbol{z}}_j$. For example, the samples $w_{1:6}^{IWAE}$ of 2 level IWAE with sample size 6 look like Eq. 16. While the samples $w_{1:2,1:3}^{DMS}$ of MS-NIC-DMS with $2 \times 3$ samples look like Eq. 17. We can see that in IWAE, we have 6 pairs of independently sampled $\tilde{\boldsymbol{y}}$ and $\tilde{\boldsymbol{z}}$, while in MS-NIC-DMS, we have 2 independent $\tilde{\boldsymbol{y}}$ and 3 independent $\tilde{\boldsymbol{z}}$, they are paired to generate 6 samples in total. Note that this is only applicable to NIC as $\tilde{\boldsymbol{y}}$ and $\tilde{\boldsymbol{z}}$ are conditionally independent given $\tilde{\boldsymbol{x}}$ due to *direct-y* trick.

$$w_{1:6}^{IWAE} = \{p(\boldsymbol{x}|\tilde{\boldsymbol{y}}_1)p(\tilde{\boldsymbol{y}}_1|\tilde{\boldsymbol{z}}_1)p(\tilde{\boldsymbol{z}}_1),$$
$$p(\boldsymbol{x}|\tilde{\boldsymbol{y}}_2)p(\tilde{\boldsymbol{y}}_2|\tilde{\boldsymbol{z}}_2)p(\tilde{\boldsymbol{z}}_2),$$
$$p(\boldsymbol{x}|\tilde{\boldsymbol{y}}_3)p(\tilde{\boldsymbol{y}}_3|\tilde{\boldsymbol{z}}_3)p(\tilde{\boldsymbol{z}}_3),$$
$$p(\boldsymbol{x}|\tilde{\boldsymbol{y}}_4)p(\tilde{\boldsymbol{y}}_4|\tilde{\boldsymbol{z}}_4)p(\tilde{\boldsymbol{z}}_4),$$
$$p(\boldsymbol{x}|\tilde{\boldsymbol{y}}_5)p(\tilde{\boldsymbol{y}}_5|\tilde{\boldsymbol{z}}_5)p(\tilde{\boldsymbol{z}}_5),$$
$$p(\boldsymbol{x}|\tilde{\boldsymbol{y}}_6)p(\tilde{\boldsymbol{y}}_6|\tilde{\boldsymbol{z}}_6)p(\tilde{\boldsymbol{z}}_6)\} \tag{16}$$

$$w_{1:2,1:3}^{DMS} = \{p(\boldsymbol{x}|\tilde{\boldsymbol{y}}_1)p(\tilde{\boldsymbol{y}}_1|\tilde{\boldsymbol{z}}_1)p(\tilde{\boldsymbol{z}}_1),$$
$$p(\boldsymbol{x}|\tilde{\boldsymbol{y}}_1)p(\tilde{\boldsymbol{y}}_1|\tilde{\boldsymbol{z}}_2)p(\tilde{\boldsymbol{z}}_2),$$
$$p(\boldsymbol{x}|\tilde{\boldsymbol{y}}_1)p(\tilde{\boldsymbol{y}}_1|\tilde{\boldsymbol{z}}_3)p(\tilde{\boldsymbol{z}}_3),$$
$$p(\boldsymbol{x}|\tilde{\boldsymbol{y}}_2)p(\tilde{\boldsymbol{y}}_2|\tilde{\boldsymbol{z}}_1)p(\tilde{\boldsymbol{z}}_1),$$
$$p(\boldsymbol{x}|\tilde{\boldsymbol{y}}_2)p(\tilde{\boldsymbol{y}}_2|\tilde{\boldsymbol{z}}_2)p(\tilde{\boldsymbol{z}}_2),$$
$$p(\boldsymbol{x}|\tilde{\boldsymbol{y}}_2)p(\tilde{\boldsymbol{y}}_2|\tilde{\boldsymbol{z}}_3)p(\tilde{\boldsymbol{z}}_3)\} \tag{17}$$

# 4 Related Work

## 4.1 Lossy Neural Image and Video Compression

Ballé et al. [2017] and Ballé et al. [2018] formulate lossy neural image compression as a variational inference problem, by interpreting the additive uniform noise (AUN) relaxed scalar quantization as a factorized uniform variational posterior. After that, the majority of sota lossy neural image compression methods adopt this formulation [Minnen et al., 2018, Minnen and Singh, 2020, Cheng et al., 2020, Guo et al., 2021a, Gao et al., 2021, He et al., 2022]. And Yang et al. [2020], Guo et al. [2021b] also require a AUN trained NIC as base. Moreover, the majority of neural video compression also adopts this formulation [Lu et al., 2019, 2020, Agustsson et al., 2020, Hu et al., 2021, Li et al., 2021], implying that MS-NIC can be extended to video compression without much pain.

Other approaches to train NIC include random rounding [Toderici et al., 2015, 2017] and straight through estimator (STE) [Theis et al., 2017]. Another promising approach is the VQ-VAE [Van Den Oord et al., 2017]. By the submission of this manuscript, one unarchived work [Zhu et al., 2022] has shown the potential of VQ-VAE in practical NIC. Our MS-NIC does not apply to the approaches mentioned in this paragraph, as the formulation of variational posterior is different.

## 4.2 Tighter Lower Bound for VAE

IWAE [Burda et al., 2016] stirs up the discussion of adopting tighter lower bound for training VAEs. However, at the first glance it is not straightforward why it might works. Cremer et al. [2018] decomposes the inference suboptimality of VAE into two parts: 1) The limited expressiveness of interence model. 2) The gap between ELBO and log likelihood. However, this gap refers to inference not training. The original IWAE paper empirically shows that IWAE can learn a richer latent representation. And Cremer et al. [2017] shows that the IWAE target converges to ELBO under the expectation of true posterior. And thus the posterior collapse is avoided.

From the information preference [Chen et al., 2017] perspective, VAE prefers to distribute information in generative distribution than autoencoding information in the latent. This preference formulates another view of posterior collapse. And it stems from the gap between ELBO and true log likelihood. There are various approaches alleviating it, including *soft free bits* [Theis et al., 2017] and *KL annealing* [Serban et al., 2017]. In our opinion, IWAE also belongs to those methods, and it is asymptotically optimal. However, we have not found many works comparing IWAE with those methods. Moreover, those approaches are rarely adopted in NIC community.

Many follow-ups of IWAE stress gradient variance reduction [Roeder et al., 2017, Tucker et al., 2018, Rainforth et al., 2018], discrete latent [Mnih and Rezende, 2016] and debiasing IWAE target [Nowozin, 2018]. Although the idea of tighter low bound training has been applied to the field of neural joint source channel coding [Choi et al., 2018, Song et al., 2020], to the best of our knowledge, no work in NIC consider it yet.

## 4.3 Multi-Sample Inference for Neural Image Compression

Theis and Ho [2021] considers the similar topic of importance weighted NIC. However, it does not consider training of NIC. Instead, it focuses on achieving IWAE target with an entropy coding technique named *softmin*, just like BB-ANS [Townsend et al., 2018] achieving ELBO. It is alluring to apply *softmin* to MS-NIC, as it closes the multiple-sample training and inference gap. However, it requires large number of samples (e.g. $4096$) to achieve slight improvement for $64 \times 64$ images. The potential sample size required for practical NIC is forbidding. Moreover, we believe the stochastic lossy encoding scheme [Agustsson and Theis, 2020] that Theis and Ho [2021] is not yet ready to be applied (See Appendix. A.7 for details).

# 5 Experimental Results

## 5.1 Experimental Settings

Following He et al. [2022], we train all the models on the largest 8000 images of ImageNet [Deng et al., 2009], followed by a downsampling according to Ballé et al. [2018]. And we use Kodak [Kodak,

1993] for evaluation. For the experiments based on Ballé et al. [2018] (include Tab. 1, Tab. 2), we follows the setting of the original paper except for the selection of $\lambda$s, For the selection of $\lambda$s, we set $\lambda \in \{0.0016, 0.0032, 0.0075, 0.015, 0.03, 0.045, 0.08\}$ as suggested in Cheng et al. [2020]. And for the experiments based on Cheng et al. [2020], we follows the setting of original paper. More detailed experimental settings can be found in Appendix. A.4.

And when comparing the R-D performance of models trained on multiple $\lambda$s, we use Bjontegaard metric (BD-Metric) and Bjontegaard bitrate (BD-BR) [Bjontegaard, 2001], which is widely applied when comparing codecs. More detailed experimental results can be found in Appendix. A.5.

Table 4: Results based on Ballé et al. [2018].

|  | PSNR | | MS-SSIM | |
|---|---|---|---|---|
|  | BD-BR (%) | BD-Metric | BD-BR (%) | BD-Metric |
| *Single-sample* | | | | |
| Baseline [Ballé et al., 2018] | 0.000 | 0.000 | 0.000 | 0.0000 |
| *Multiple-sample* | | | | |
| IWAE [Burda et al., 2016] | 64.23 | -2.318 | 68.67 | -0.01648 |
| MS-NIC-MIX | -3.847 | 0.1877 | -4.743 | 0.001618 |
| MS-NIC-DMS | -4.929 | 0.2405 | -5.617 | 0.001976 |

Table 5: Results based on Cheng et al. [2020]. The BD Metrics of IWAE can not be computed as its R-D is not monotonously increasing.

|  | PSNR | | MS-SSIM | |
|---|---|---|---|---|
|  | BD-BR (%) | BD-Metric | BD-BR (%) | BD-Metric |
| *Single-sample* | | | | |
| Baseline [Cheng et al., 2020] | 0.0000 | 0.0000 | 0.0000 | 0.0000 |
| *Multiple-sample* | | | | |
| IWAE [Burda et al., 2016] | - | - | - | - |
| MS-NIC-MIX | -1.852 | 0.0805 | 2.238 | -0.0006764 |
| MS-NIC-DMS | -2.378 | 0.1046 | 1.998 | -0.0006054 |

## 5.2  R-D Performance

We evaluate the performance of MS-NIC-MIX and MS-NIC-DMS based on sota NIC methods [Ballé et al., 2018, Cheng et al., 2020]. Empirically, we find that MS-NIC-MIX works best with sample size 8, and MS-NIC-DMS with sample size 16. The experimental results on sample size selection can be found in Appendix. A.3. Without special mention, we set the sample size of MS-NIC-MIX to 8 and MS-NIC-DMS to 16.

For Ballé et al. [2018], MS-NIC-MIX saves around $4\%$ of bitrate compared with single-sample baseline (See Tab. 4). And MS-NIC-DMS saves around $5\%$ of bitrate. On the other hand, the original IWAE suffers performance decay as it is not compatible with *direct-y* trick. For Cheng et al. [2020], we find that both MS-NIC-MIX and NS-NIC-DMS suppress baseline in PSNR. However, it is not as evident as Ballé et al. [2018]. Moreover, the MS-SSIM is slightly lower than the baseline. This is probably due to the auto-regressive context model. Besides, the original IWAE without *direct-y* trick suffers from severe performance decay in both cases. The BD metric of IWAE on Cheng et al. [2020] can not be computed as its R-D is not monotonous increasing, we refer interested readers to Appendix. A.5 for details.

## 5.3  Latent Space Representation of MS-NIC

To better understand the latent learned by MS-NIC, we evaluate the variance and coefficient of variation (Cov) of per-dimension latent distribution mean parameter $\boldsymbol{y}^{(i)}, \boldsymbol{z}^{(i)}$, with regard to input

distribution $p(\boldsymbol{x})$. As we are also interested in the discrete representation, we provide statistics of rounded mean $\bar{\boldsymbol{y}}^{(i)}, \bar{\boldsymbol{z}}^{(i)}$. These metrics show how much do latents vary when input changes, and a large variation in latents means that there are useful information encoded. A really small variation indicates that the latent is "dead" in that dimension.

As shown in Tab. 10 of Appendix. A.6, the latent of multiple-sample approaches has higher variance than those of single-sample approach. Moreover, the $\text{Cov}(\boldsymbol{y})$ of multiple-sample approaches is around $4-5$ times higher than single-sample approach. Although the $\text{Cov}(\boldsymbol{z})$ of multiple-sample approaches is around $2$ times lower, the main contributor of image reconstruction is $\boldsymbol{y}$, and $\boldsymbol{z}$ only serves to predict $\boldsymbol{y}$'s distribution. Similar trend can be concluded from quantized latents $\bar{\boldsymbol{y}}, \bar{\boldsymbol{z}}$. From the variance and Cov perspective, the latent learned by MS-NIC is richer than single-sample approach. It is also noteworthy that although the variance and Cov of $\boldsymbol{y}, \bar{\boldsymbol{y}}$ of MS-NIC is significantly higher than single-sample approach, the bpp only varies slightly.

Table 6: The average of per-dimension latent variance and Cov across Kodak test images. The model is trained with $\lambda = 0.015$.

| Method | Var(#) | | Cov(#) | | bpp of # | |
|---|---|---|---|---|---|---|
| | $\bar{y}$ | $\bar{z}$ | $\bar{y}$ | $\bar{z}$ | $\bar{y}$ | $\bar{z}$ |
| *Single-sample* | | | | | | |
| Ballé et al. [2018] | 1.499 | 0.3255 | 19.70 | 9.944 | 0.5136 | 0.01342 |
| *Multiple-sample* | | | | | | |
| MS-NIC-MIX | 1.906 | 0.7594 | 111.1 | 7.425 | 0.5108 | 0.01521 |
| MS-NIC-DMS | 1.919 | 0.7648 | 95.51 | 7.243 | 0.5092 | 0.01634 |

# 6  Limitation & Discussion

A major limitation of our method is that the improvement in R-D performance is marginal, especially when based on Cheng et al. [2020]. Moreover, evaluations on more recent sota methods are also helpful to strengthen the claims of this paper. In general, we think that the performance improvement of our approach is bounded by how severe the posterior collapse is in neural image compression. We measure the variance in latent dimension according to data in Fig. A.6. And from that figure it might be observed that the major divergence of IWAE and VAE happens when the variance is very small. And for the area where variance is reasonably large, the gain of IWAE is not that large. This probably indicates that the posterior collapse in neural image compression is only alleviated to a limited extend.

See more discussion in why the result on Cheng et al. [2020] is negative in Appendix. A.9

# 7  Conclusion

In this paper we propose MS-NIC, a multiple-sample importance weighted target for training NIC. It improves sota NIC methods and learns richer latent representation. A known limitation is that its R-D performance improvement is limited when applied to models with spatial context models (e.g. Cheng et al. [2020]). Despite the somewhat negative result, this paper provides insights to the training of NIC models from VAE perspective. Further work could consider improving the performance and extend it into neural video compression.

## Acknowledgments and Disclosure of Funding

This work is supported by SenseTime Research. The content is solely the responsibility of the authors and does not necessarily represent the official views of SenseTime Research.

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
