# A   Appendix

## A.1   Plate Notations of Generative Models

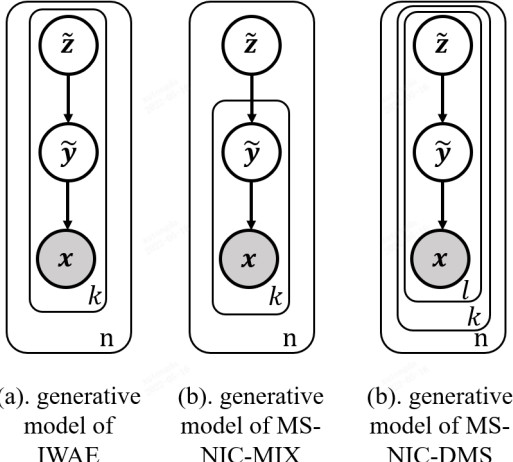

(a). generative model of IWAE    (b). generative model of MS-NIC-MIX    (b). generative model of MS-NIC-DMS

Figure 2: The generative model of Fig. 1 (d)-(f). The generative models show how we compute data likelihood for multiple-sample approaches, not how the image is actually generated in nature. For clarity, we omit the parameters.

Fig. A.1 shows the generative models of Fig. 1 (d)-(f). Note that we have only one unique sample of $\boldsymbol{x}$ inside the $n$ plate, but it is repeated $k$ times for IWAE, MS-NIC-MIX and $k \times l$ times for MS-NIC-DMS. Similarly, k samples of $\tilde{\boldsymbol{y}}$ is repeated $l$ times and $l$ samples of $\tilde{\boldsymbol{z}}$ is repeated $k$ times for MS-NIC-DMS.

## A.2   Proof on the Properties of MS-NIC-MIX and MS-NIC-DMS

In this section, we add the $q(\tilde{\boldsymbol{z}}|\tilde{\boldsymbol{y}}), q(\tilde{\boldsymbol{y}}|\boldsymbol{x})$ back to equations for clarity of the proof. This makes the notations slightly different from Eq. 12 and Eq. 14. Note that divide by $q(\tilde{\boldsymbol{z}}|\tilde{\boldsymbol{y}}), q(\tilde{\boldsymbol{y}}|\boldsymbol{x})$ does not effect the value of equation, and add $\log q(\tilde{\boldsymbol{z}}|\tilde{\boldsymbol{y}}), \log q(\tilde{\boldsymbol{y}}|\boldsymbol{x})$ does not effect the value of equation.

For MS-NIC-MIX to be a reasonably better approach to apply over [Ballé et al., 2018], we show that $\mathcal{L}_k^{MIX}$ satisfies following properties:

1. $\log p(x) \geq \mathcal{L}_k^{MIX}$
2. $\mathcal{L}_k^{MIX} \geq \mathcal{L}_m^{MIX}$ for $k \geq m$

We can show 1. $\log p(x) \geq \mathcal{L}_k^{MIX}$ by applying Jensen's inequality twice:

$$
\begin{aligned}
\mathcal{L}_k^{MIX} &= \mathbb{E}_{q_\phi(\tilde{\boldsymbol{z}}|\boldsymbol{x})}[\mathbb{E}_{q_\phi(\tilde{\boldsymbol{y}}_{1:k}|\boldsymbol{x})}[\log \tfrac{1}{k}\sum_i^k \tfrac{p(\boldsymbol{x}|\tilde{\boldsymbol{y}}_i)p(\tilde{\boldsymbol{y}}_i|\tilde{\boldsymbol{z}})}{q(\tilde{\boldsymbol{y}}_i|\boldsymbol{x})}|\tilde{\boldsymbol{z}}] + \log p(\tilde{\boldsymbol{z}}) - \log q(\tilde{\boldsymbol{z}}|\boldsymbol{x})] \\
&\leq \mathbb{E}_{q_\phi(\tilde{\boldsymbol{z}}|\boldsymbol{x})}[\log(\tfrac{1}{k}\sum_i^k \mathbb{E}_{q_\phi(\tilde{\boldsymbol{y}}_{1:k}|\boldsymbol{x})}[\tfrac{p(\boldsymbol{x}|\tilde{\boldsymbol{y}}_i)p(\tilde{\boldsymbol{y}}_i|\tilde{\boldsymbol{z}})}{q(\tilde{\boldsymbol{y}}_i|\boldsymbol{x})}|\tilde{\boldsymbol{z}}]) + \log p(\tilde{\boldsymbol{z}}) - \log q(\tilde{\boldsymbol{z}}|\boldsymbol{x})] \\
&= \mathbb{E}_{q_\phi(\tilde{\boldsymbol{z}}|\boldsymbol{x})}[\log p(\boldsymbol{x}|\tilde{\boldsymbol{z}}) + \log p(\tilde{\boldsymbol{z}}) - \log q(\tilde{\boldsymbol{z}}|\boldsymbol{x})] \\
&\leq \log(\mathbb{E}_{q_\phi(\tilde{\boldsymbol{z}}|\boldsymbol{x})}[\tfrac{p(\boldsymbol{x}|\tilde{\boldsymbol{z}})p(\tilde{\boldsymbol{z}})}{q(\tilde{\boldsymbol{z}}|\boldsymbol{x})}]) \\
&= \log p(x)
\end{aligned}
\tag{18}
$$

We can show 2. $\mathcal{L}_k^{MIX} \geq \mathcal{L}_m^{MIX}$ for $k \geq m$ by borrowing the Theorem 1 from IWAE paper:

$$
k \geq m \Rightarrow \mathbb{E}_{q(\boldsymbol{h}_i|\boldsymbol{x})}[\log \tfrac{1}{k}\sum_i^k \tfrac{p(\boldsymbol{h}_i|\boldsymbol{x})p(\boldsymbol{h}_i)}{q(\boldsymbol{h}_i|\boldsymbol{x})}] \geq \mathbb{E}_{q(\boldsymbol{h}_i|\boldsymbol{x})}[\log \tfrac{1}{m}\sum_i^m \tfrac{p(\boldsymbol{h}_i|\boldsymbol{x})p(\boldsymbol{h}_i)}{q(\boldsymbol{h}_i|\boldsymbol{x})}]
\tag{19}
$$

Applying Eq. 19 to the internal part of $\mathcal{L}_k^{MIX}$, when $k \geq m$, we have:

$$
\begin{aligned}
\mathcal{L}_k^{MIX} &= \mathbb{E}_{q_\phi(\tilde{\boldsymbol{z}}|\boldsymbol{x})}[\mathbb{E}_{q_\phi(\tilde{\boldsymbol{y}}_{1:k}|\boldsymbol{x})}[\log \tfrac{1}{k}\sum_i^k \tfrac{p(\boldsymbol{x}|\tilde{\boldsymbol{y}}_i)p(\tilde{\boldsymbol{y}}_i|\tilde{\boldsymbol{z}})}{q(\tilde{\boldsymbol{y}}_i|\boldsymbol{x})}|\tilde{\boldsymbol{z}}] + \log p(\tilde{\boldsymbol{z}}) - \log q(\tilde{\boldsymbol{z}}|\boldsymbol{x})] \\
&\geq \mathbb{E}_{q_\phi(\tilde{\boldsymbol{z}}|\boldsymbol{x})}[\mathbb{E}_{q_\phi(\tilde{\boldsymbol{y}}_{1:m}|\boldsymbol{x})}[\log \tfrac{1}{m}\sum_i^m \tfrac{p(\boldsymbol{x}|\tilde{\boldsymbol{y}}_i)p(\tilde{\boldsymbol{y}}_i|\tilde{\boldsymbol{z}})}{q(\tilde{\boldsymbol{y}}_i|\boldsymbol{x})}|\tilde{\boldsymbol{z}}] + \log p(\tilde{\boldsymbol{z}}) - \log q(\tilde{\boldsymbol{z}}|\boldsymbol{x})] \\
&= \mathcal{L}_m^{MIX}
\end{aligned}
\tag{20}
$$

For MS-NIC-DMS to be a reasonably better approach to apply over [Ballé et al., 2018] and MS-NIC-MIX, we show that $\mathcal{L}_{k,l}^{DMS}$ statisfies following properties:

1. $\log p(x) \geq \mathcal{L}_{k,l}^{DMS}$

2. $\mathcal{L}_{k,l}^{DMS} \geq \mathcal{L}_{m,n}^{DMS}$ for $k \geq m, l \geq n$

3. $\mathcal{L}_{k,l}^{DMS} \geq \mathcal{L}_k^{MIX}$

4. $\mathcal{L}_{k,l}^{DMS} \to \log p(x)$ as $k,l \to \infty$, under the assumption that $\log \frac{p(\boldsymbol{x}|\tilde{\boldsymbol{y}}_i)p(\tilde{\boldsymbol{y}}_i|\tilde{\boldsymbol{z}}_j)}{q(\tilde{\boldsymbol{y}}_i|\boldsymbol{x})}$ and $\log \frac{p(\boldsymbol{x}|\tilde{\boldsymbol{z}}_j)p(\tilde{\boldsymbol{z}}_j)}{q(\tilde{\boldsymbol{z}}_j|\boldsymbol{x})}$ are bounded.

Similar to MS-NIC-DMS, we can show $1.\log p(x) \geq \mathcal{L}_{k,l}^{DMS}$ by applying Jensen's inequality twice:

$$
\begin{aligned}
\mathcal{L}_{k,l}^{DMS} &= \mathbb{E}_{q_\phi(\tilde{\boldsymbol{z}}_{1:l}|\boldsymbol{x})}[\log \tfrac{1}{l}\sum_j^l \exp\left(\mathbb{E}_{q_\phi(\tilde{\boldsymbol{y}}_{1:k}|\boldsymbol{x})}[\log \tfrac{1}{k}\sum_i^k \tfrac{p(\boldsymbol{x}|\tilde{\boldsymbol{y}}_i)p(\tilde{\boldsymbol{y}}_i|\tilde{\boldsymbol{z}}_j)}{q(\tilde{\boldsymbol{y}}_i|\boldsymbol{x})}|\tilde{\boldsymbol{z}}_j]\right)p(\tilde{\boldsymbol{z}}_j)/q(\tilde{\boldsymbol{z}}_j|\boldsymbol{x})] \\
&\leq \mathbb{E}_{q_\phi(\tilde{\boldsymbol{z}}_{1:l}|\boldsymbol{x})}[\log \tfrac{1}{l}\sum_j^l \exp\log(\tfrac{1}{k}\sum_i^k\mathbb{E}_{q_\phi(\tilde{\boldsymbol{y}}_{1:k}|\boldsymbol{x})}[p(\boldsymbol{x}|\tilde{\boldsymbol{y}}_i)p(\tilde{\boldsymbol{y}}_i|\tilde{\boldsymbol{z}}_j)|\tilde{\boldsymbol{z}}_j])p(\tilde{\boldsymbol{z}}_j)/q(\tilde{\boldsymbol{z}}_j|\boldsymbol{x})] \\
&= \mathbb{E}_{q_\phi(\tilde{\boldsymbol{z}}_{1:l}|\boldsymbol{x})}[\log \tfrac{1}{l}\sum_j^l \tfrac{p(\boldsymbol{x}|\tilde{\boldsymbol{z}}_j)p(\tilde{\boldsymbol{z}}_j)}{q(\tilde{\boldsymbol{z}}_j|\boldsymbol{x})}] \\
&\leq \log \tfrac{1}{l}\sum_j^l\mathbb{E}_{q_\phi(\tilde{\boldsymbol{z}}_{1:l}|\boldsymbol{x})}[\tfrac{p(\boldsymbol{x}|\tilde{\boldsymbol{z}}_j)p(\tilde{\boldsymbol{z}}_j)}{q(\tilde{\boldsymbol{z}}_j|\boldsymbol{x})}] \\
&= \log p(\boldsymbol{x})
\end{aligned}
\tag{21}
$$

Also similar to MS-NIC-MIX, we can borrow conclusion from IWAE (Eq. 19) and apply it twice to show $2.\ \mathcal{L}_{k,l}^{DMS} \geq \mathcal{L}_{m,n}^{DMS}$ for $k \geq m, l \geq n$:

$$
\begin{aligned}
\mathcal{L}_{k,l}^{DMS} &= \mathbb{E}_{q_\phi(\tilde{\boldsymbol{z}}_{1:l}|\boldsymbol{x})}[\log \tfrac{1}{l}\sum_j^l \exp\left(\mathbb{E}_{q_\phi(\tilde{\boldsymbol{y}}_{1:k}|\boldsymbol{x})}[\log \tfrac{1}{k}\sum_i^k \tfrac{p(\boldsymbol{x}|\tilde{\boldsymbol{y}}_i)p(\tilde{\boldsymbol{y}}_i|\tilde{\boldsymbol{z}}_j)}{q(\tilde{\boldsymbol{y}}_i|\boldsymbol{x})}|\tilde{\boldsymbol{z}}_j]\right)p(\tilde{\boldsymbol{z}}_j)/q(\tilde{\boldsymbol{z}}_j|\boldsymbol{x})] \\
&\geq \mathbb{E}_{q_\phi(\tilde{\boldsymbol{z}}_{1:l}|\boldsymbol{x})}[\log \tfrac{1}{l}\sum_j^l \exp\left(\mathbb{E}_{q_\phi(\tilde{\boldsymbol{y}}_{1:m}|\boldsymbol{x})}[\log \tfrac{1}{m}\sum_i^m \tfrac{p(\boldsymbol{x}|\tilde{\boldsymbol{y}}_i)p(\tilde{\boldsymbol{y}}_i|\tilde{\boldsymbol{z}}_j)}{q(\tilde{\boldsymbol{y}}_i|\boldsymbol{x})}|\tilde{\boldsymbol{z}}_j]\right)p(\tilde{\boldsymbol{z}}_j)/q(\tilde{\boldsymbol{z}}_j|\boldsymbol{x})] \\
&\geq \mathbb{E}_{q_\phi(\tilde{\boldsymbol{z}}_{1:n}|\boldsymbol{x})}[\log \tfrac{1}{n}\sum_j^n \exp\left(\mathbb{E}_{q_\phi(\tilde{\boldsymbol{y}}_{1:m}|\boldsymbol{x})}[\log \tfrac{1}{m}\sum_i^m \tfrac{p(\boldsymbol{x}|\tilde{\boldsymbol{y}}_i)p(\tilde{\boldsymbol{y}}_i|\tilde{\boldsymbol{z}}_j)}{q(\tilde{\boldsymbol{y}}_i|\boldsymbol{x})}|\tilde{\boldsymbol{z}}_j]\right)p(\tilde{\boldsymbol{z}}_j)/q(\tilde{\boldsymbol{z}}_j|\boldsymbol{x})] \\
&= \mathcal{L}_{m,n}^{DMS}
\end{aligned}
\tag{22}
$$

With $2.\ \mathcal{L}_{k,l}^{DMS} \geq \mathcal{L}_{m,n}^{DMS}$ for $k \geq m, l \geq n$ holds, we can show $3.\ \mathcal{L}_{k,l}^{DMS} \geq \mathcal{L}_k^{MIX}$ immediately as $\mathcal{L}_{k,l}^{DMS} \geq \mathcal{L}_{k,1}^{DMS} = \mathcal{L}_k^{MIX}$.

To show $4.\ \mathcal{L}_{k,l}^{DMS} \to \log p(x)$ as $k,l \to \infty$, we first define intermediate variables $W_k, \tilde{M}_{k,l}, M_{k,l}$:

$$W_k = \frac{1}{k}\sum_i^k \frac{p(\boldsymbol{x}|\tilde{\boldsymbol{y}}_i)p(\tilde{\boldsymbol{y}}_i|\tilde{\boldsymbol{z}}_j)}{q(\tilde{\boldsymbol{y}}_i|\boldsymbol{x})}$$

$$\tilde{M}_{k,l} = \frac{1}{l}\sum_j^l \frac{p(\boldsymbol{x}|\tilde{\boldsymbol{z}}_j)p(\tilde{\boldsymbol{z}}_j)}{q(\tilde{\boldsymbol{z}}_j|\boldsymbol{x})} \qquad (23)$$

$$M_{k,l} = \frac{1}{l}\sum_j^l \exp\left(\mathbb{E}_{q_\phi(\tilde{\boldsymbol{y}}_{1:k}|\boldsymbol{x})}[\log W_k|\tilde{\boldsymbol{z}}_j]\right)p(\tilde{\boldsymbol{z}}_j)/q(\tilde{\boldsymbol{z}}_j|\boldsymbol{x})$$

Under the assumption that $\log p(\boldsymbol{x}|\tilde{\boldsymbol{y}}_i)p(\tilde{\boldsymbol{y}}_i|\tilde{\boldsymbol{z}}_j)/q(\tilde{\boldsymbol{y}}_i|\boldsymbol{x})$ is bounded, from the strong law of large number, we have $W_k \xrightarrow{a.s.} p(\boldsymbol{x}|\tilde{\boldsymbol{z}}_j)$ (Eq. 24). Then we have $\mathbb{E}[\log W_k|\tilde{\boldsymbol{z}}_j] \to \log p(\boldsymbol{x}|\tilde{\boldsymbol{z}}_j)$.

$$W_k \xrightarrow{a.s.} \mathbb{E}_{q(\tilde{\boldsymbol{y}}_i|\boldsymbol{x})}\left[\frac{p(\boldsymbol{x}|\tilde{\boldsymbol{y}}_i)p(\tilde{\boldsymbol{y}}_i|\tilde{\boldsymbol{z}}_j)}{q(\tilde{\boldsymbol{y}}_i|\boldsymbol{x})}|\tilde{\boldsymbol{z}}_j\right] = \int q(\tilde{\boldsymbol{y}}_i|\boldsymbol{x})\frac{p(\boldsymbol{x}|\tilde{\boldsymbol{y}}_i)p(\tilde{\boldsymbol{y}}_i|\tilde{\boldsymbol{z}}_j)}{q(\tilde{\boldsymbol{y}}_i|\boldsymbol{x})}d\tilde{\boldsymbol{y}}_i = p(\boldsymbol{x}|\tilde{\boldsymbol{z}}_j) \qquad (24)$$

Moreover, as $\mathbb{E}[\log W_k|\tilde{\boldsymbol{z}}_j] \to \log p(\boldsymbol{x}|\tilde{\boldsymbol{z}}_j)$, we have $M_{k,l} \to \tilde{M}_{k,l}$. This means that $\forall \epsilon > 0, \exists k, l, s.t. |M_{k,l} - \tilde{M}_{k,l}| < \epsilon$. And thus we have $|\mathbb{E}[M_{k,l}] - \mathbb{E}[\tilde{M}_{k,l}]| \le \mathbb{E}[|M_{k,l} - \tilde{M}_{k,l}|] < \epsilon \to 0$. Then we have $|E[M_{k,l}] - p(\boldsymbol{x})| \le |E[M_{k,l}] - E[\tilde{M}_{k,l}]| + |E[\tilde{M}_{k,l}] - p(\boldsymbol{x})| \to 0$, and thus $E[M_{k,l}] \to p(\boldsymbol{x})$. Finally we have $\mathbb{E}[\log M_{k,l}] = L_{k,l}^{DMS} \to \log p(\boldsymbol{x})$.

### A.3  Effects of Sample Size

When comparing the R-D performance of models trained with a single $\lambda$, we use R-D cost as our metric. The R-D cost is simply computed as bpp $+\lambda$ MSE, where bpp is a short of bits-per-pixel, and MSE is a short of mean square error. The lower the R-D cost is, the better the R-D performance is. Another way to interpret R-D cost is to view it as the ELBO with constant offset. Then the $\lambda$ MSE is connected to the log likelihood of a Gaussian distribution whose mean is the output of decoder and sigma is determined by $\lambda$. Note that R-D cost is only comparable when $\lambda$ is the same.

Tab. 7 shows the effect of sample size to MS-NIC. Moreover, we compare the naïve increase of batch size versus multiple importance weighted samples. As shown by the table, increasing the batch size $\times 3 - 16$ only slightly affects the R-D cost (from $1.017$ to $1.013$). However, the MS-NIC-MIX can achieve R-D cost of $0.9988$ with sample size 8, and MS-NIC-DMS can achieve $0.9954$ with sample size 16. This means that MS-NIC is effective over the baseline and vanilla batch size increases. It is also noteworthy that we have not observed inference model training failure as sample size increase. While MS-NIC also suffers from gradient SNR vanishing problem, a sample size of 16 is probably not large enough to make it evident. Limited by computational power, we can not raise sample size by several magnitudes as [Rainforth et al., 2018] does with small model.

### A.4  Detailed Experimental Settings

All the experiments are conducted on a computer with Intel(R) Xeon(R) CPU E5-2620 v4 @ 2.10GHz and $8\times$ Nvidia(R) TitanXp. All the training scripts are implemented with Pytorch 1.7 and CUDA 9.0. For experiments with single-sample, we adopt Adam optimizer with $\beta_1 = 0.90, \beta_2 = 0.95, lr = 1e^{-4}$. For experiments with multiple-sample/big batch, we scale $lr$ linearly with sample size. All the models are trained for 2000 epochs with the settings in Sec. 5.1. For first 200 epochs, we adopt cosine annealing [Loshchilov and Hutter, 2016] to schedule learning rate. It takes around $1 - 2$ days to train models based on Ballé et al. [2018], and $3 - 5$ days to train models on Cheng et al. [2020]. Note that our multiple-sample approaches' training time does not scale linearly with sample size, as we perform sampling on posterior, and the variational encoder only computes parameter of posterior parameters once. Further, we provide the pytorch style sudo code for implementation guidance of MS-NIC-MIX and MS-NIC-DMS.

```
import torch
from torch.nn import functional as F

def IWAELoss(minus_elbo):
    '''
    args
    ----
    minus_elbo: tensor, [b, k], which is R + \lambda D
```

Table 7: Effect of sample size in MS-NIC.

| | Sample/Batch Size | bpp | MSE | PSNR (db) | R-D cost |
|---|---|---|---|---|---|
| Baseline [Ballé et al., 2018] | - | 0.5273 | 32.61 | 33.28 | 1.017 |
| Baseline-BigBatch | ×3 | 0.5308 | 32.51 | 33.31 | 1.018 |
| | ×5 | 0.5285 | 32.51 | 33.30 | 1.016 |
| | ×8 | 0.5279 | 32.37 | 33.34 | 1.013 |
| | ×16 | 0.5321 | 32.12 | 33.38 | 1.014 |
| IWAE [Burda et al., 2016] | 3 | 0.9128 | 32.46 | 33.28 | 1.400 |
| | 5 | 0.7903 | 31.73 | 33.40 | 1.266 |
| | 8 | 0.9477 | 31.48 | 33.44 | 1.420 |
| | 16 | 1.273 | 31.69 | 33.40 | 1.748 |
| MS-NIC-MIX | 3 | 0.5238 | 31.80 | 33.40 | 1.000 |
| | 5 | 0.5259 | 31.84 | 33.38 | 1.003 |
| | 8 | 0.5260 | 31.52 | 33.44 | 0.9988 |
| | 16 | 0.5256 | 32.48 | 33.29 | 1.013 |
| MS-NIC-DMS | 3, 3 | 0.5247 | 32.39 | 33.30 | 1.010 |
| | 5, 5 | 0.5230 | 31.84 | 33.39 | 1.001 |
| | 8, 8 | 0.5255 | 31.55 | 33.43 | 0.9989 |
| | 16, 16 | 0.5249 | 31.38 | 33.46 | 0.9954 |

```
    return
    ------
    local iwae loss
    '''
    # this is the minus ELBO related to y part,
    # to get the real ELBO:
    log_weights = - minus_elbo.detach()
    # no gradient given to weights
    weights = F.softmax(log_weights, dim=1) # B, K
    loss_b = torch.sum(minus_elbo * weights, dim=1, keepdim=False)
    loss_iwae = torch.mean(loss_b)
    return loss_iwae

def DMSLoss(x, x_hat, y_likelihood, z_likelihood, lam):
    '''
    args
    ----
    x: original image: [b, c, h, w]
    x_hat: reconstructed image: [b, k, c, h, w], k is the
        number of samples
    y_likelihood: [b, 192/320, h//8, w//8, k^2],
        as original paper of [Balle et al. 2018], the number of
        channels 192/320 is determined by lambda, k^2 is the
        number of samples in DMS setting, with MS-NIC-MIX,
        this k^2 is k
    z_likelihood: [b, 128/192, h//64, w//64, k], as original
        paper of [Balle et al. 2018], the number of channels
        128/192 is determined by lambda,
        k is the number of samples

    return
    ------
    total iwae loss
    '''
    b, c, h, w = x.shape
    k = x_hat.shape[0] // x.shape[0]
    x = torch.repeat_interleave(x, repeats=k, dim=0)
```

```
x = x.reshape(b, k, c, h, w)
x_hat = x_hat.reshape(b, k, c, h, w)
d_loss = torch.mean(lam * 65025 * (x - x_hat)**2, dim=(2,3,4),
                    keepdim=False)
yz_loss = -torch.sum(torch.log2(y_likelihood), dim=(1,2,3)).\
          reshape(b, -1) / (h * w)
z_loss = -torch.sum(torch.log2(z_likelihood), dim=(1,2,3)).\
          reshape(b, -1) / (h * w)
local_d = IWAELoss(d_loss)
local_yz = IWAELoss(yz_loss)
local_z = IWAELoss(z_loss)
loss_total = local_d + local_yz + local_z

return loss_total
```

## A.5 Detailed Experimental Results

In this section we present more detailed experimental results in Tab. 8 and Tab. 5. Note that without *direct-y* trick, the IWAE for Cheng et al. [2020] totally fails and we can not produce a valid BD metric from it.

Table 8: Detailed results based on [Ballé et al., 2018].

|  | $\lambda$ | bpp | MSE | PSNR (db) | MS-SSIM |
|---|---|---|---|---|---|
| Baseline [Ballé et al., 2018] | 0.0016 | 0.1205 | 138.4 | 27.23 | 0.9111 |
|  | 0.0032 | 0.1990 | 91.52 | 28.95 | 0.9384 |
|  | 0.0075 | 0.3492 | 52.68 | 31.28 | 0.9624 |
|  | 0.015 | 0.5270 | 32.78 | 33.28 | 0.9766 |
|  | 0.03 | 0.7626 | 19.90 | 35.37 | 0.9847 |
|  | 0.045 | 0.9249 | 15.69 | 36.39 | 0.9883 |
|  | 0.08 | 1.211 | 10.04 | 38.27 | 0.9919 |
| IWAE [Burda et al., 2016] | 0.0016 | 0.2559 | 144.7 | 27.12 | 0.9134 |
|  | 0.0032 | 0.3478 | 90.65 | 29.03 | 0.9389 |
|  | 0.0075 | 0.5931 | 51.40 | 31.38 | 0.9642 |
|  | 0.015 | 0.7902 | 31.73 | 33.40 | 0.9765 |
|  | 0.03 | 1.135 | 19.41 | 35.47 | 0.9850 |
|  | 0.045 | 1.886 | 14.70 | 36.65 | 0.9885 |
|  | 0.08 | 1.753 | 9.898 | 38.32 | 0.9919 |
| MS-NIC-MIX | 0.0016 | 0.1132 | 146.4 | 27.08 | 0.9121 |
|  | 0.0032 | 0.1967 | 88.44 | 29.15 | 0.9409 |
|  | 0.0075 | 0.3496 | 51.27 | 31.39 | 0.9632 |
|  | 0.015 | 0.5260 | 31.52 | 33.43 | 0.9773 |
|  | 0.03 | 0.7591 | 19.33 | 35.49 | 0.9851 |
|  | 0.045 | 0.9248 | 14.43 | 36.72 | 0.9885 |
|  | 0.08 | 1.201 | 9.694 | 38.40 | 0.9919 |
| MS-NIC-DMS | 0.0016 | 0.1173 | 135.1 | 27.38 | 0.9145 |
|  | 0.0032 | 0.1967 | 86.07 | 29.26 | 0.9413 |
|  | 0.0075 | 0.3495 | 49.98 | 31.51 | 0.9647 |
|  | 0.015 | 0.5250 | 31.36 | 33.46 | 0.9771 |
|  | 0.03 | 0.7546 | 19.81 | 35.37 | 0.9846 |
|  | 0.045 | 0.9220 | 14.74 | 36.61 | 0.9883 |
|  | 0.08 | 1.196 | 9.637 | 38.43 | 0.9920 |

## A.6 Distribution of Latent Variance

We show the histogram of latent variance in log space in Fig. A.6. From the histogram we can observe that for latent $y$, the variance distribution of two MS-NIC approaches is similar and single-sample

Table 9: Detailed results based on [Cheng et al., 2020].

|  | $\lambda$ | bpp | MSE | PSNR (db) | MS-SSIM |
|---|---|---|---|---|---|
| Baseline [Cheng et al., 2020] | 0.0016 | 0.1205 | 138.4 | 27.23 | 0.9111 |
|  | 0.0032 | 0.1990 | 91.52 | 28.95 | 0.9384 |
|  | 0.0075 | 0.3492 | 52.68 | 31.28 | 0.9624 |
|  | 0.015 | 0.5270 | 32.78 | 33.28 | 0.9766 |
|  | 0.03 | 0.6424 | 19.48 | 35.54 | 0.9855 |
|  | 0.045 | 0.7846 | 15.48 | 36.53 | 0.9885 |
|  | 0.08 | 1.026 | 11.41 | 37.87 | 0.9916 |
| IWAE [Burda et al., 2016] | 0.0016 | 3.226 | 109.1 | 28.32 | 0.9182 |
|  | 0.0032 | 3.407 | 78.07 | 29.74 | 0.9414 |
|  | 0.0075 | 3.555 | 47.19 | 31.84 | 0.9652 |
|  | 0.015 | 3.445 | 31.84 | 33.56 | 0.9779 |
|  | 0.03 | 3.534 | 23.66 | 34.92 | 0.9849 |
|  | 0.045 | 3.545 | 20.63 | 35.59 | 0.9878 |
|  | 0.08 | 3.157 | 16.40 | 36.62 | 0.9908 |
| MS-NIC-MIX | 0.0016 | 0.1068 | 109.7 | 28.30 | 0.9171 |
|  | 0.0032 | 0.1636 | 78.07 | 29.71 | 0.9404 |
|  | 0.0075 | 0.2861 | 47.29 | 31.85 | 0.9651 |
|  | 0.015 | 0.4309 | 31.88 | 33.55 | 0.9777 |
|  | 0.03 | 0.6586 | 19.11 | 35.60 | 0.9853 |
|  | 0.045 | 0.8007 | 14.65 | 36.76 | 0.9889 |
|  | 0.08 | 1.034 | 10.93 | 38.00 | 0.9916 |
| MS-NIC-DMS | 0.0016 | 0.1043 | 109.8 | 28.30 | 0.9163 |
|  | 0.0032 | 0.1644 | 77.44 | 29.74 | 0.9412 |
|  | 0.0075 | 0.2849 | 47.30 | 31.85 | 0.9656 |
|  | 0.015 | 0.4306 | 32.22 | 33.51 | 0.9775 |
|  | 0.03 | 0.6432 | 18.94 | 35.65 | 0.9856 |
|  | 0.045 | 0.7926 | 14.84 | 36.67 | 0.9886 |
|  | 0.08 | 1.039 | 10.48 | 38.18 | 0.9918 |

approach is quite different. MS-NIC has more latent dimensions that have high variance (the right mode), and less with low variance (the left mode). Moreover, the low variance mode of MS-NIC has less variance than single-sample approach, which indicates that MS-NIC does a better job in separating active and inactive latent dimensions. Similarly, the low variance mode of $\tilde{z}$ in MS-NIC approaches is lower than single-sample approach.

Table 10: The average of per-dimension latent variance and Cov across Kodak test images. The model is trained with $\lambda = 0.015$.

|  | Var(#) | | Cov(#) | |
|---|---|---|---|---|
| Method | $y$ | $z$ | $y$ | $z$ |
| *Single-sample* |  |  |  |  |
| Ballé et al. [2018] | 1.512 | 0.3356 | 20.36 | 14.48 |
| *Multiple-sample* |  |  |  |  |
| MS-NIC-MIX | 1.909 | 0.7705 | 114.4 | 7.234 |
| MS-NIC-DMS | 1.908 | 0.7522 | 93.67 | 7.024 |

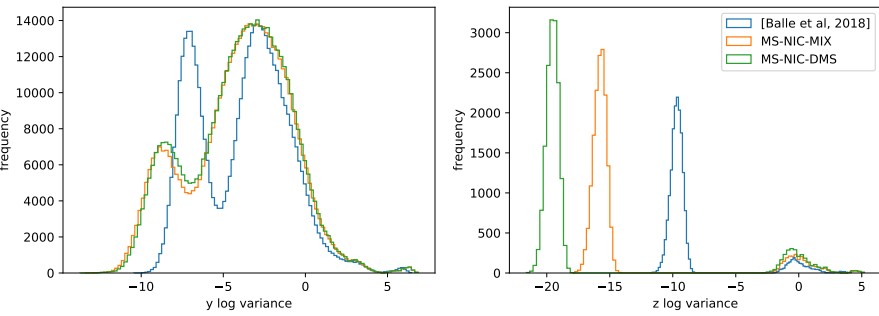

Figure 3: The histogram of log space per-dimension latent variance across Kodak test images. The model is trained with $\lambda = 0.015$.

## A.7 Tighter ELBO for Inference Time

### A.7.1 Inference time ELBO and Softmin Coding [Theis and Ho, 2021]

The inference time tighter ELBO is another under-explored issue. In fact, the training time tighter ELBO and inference time ELBO is independent. We can train a model with tighter ELBO, infer with single sample ELBO. Or we can also conduct multiple sample infer on a model trained with single sample. The general idea is:

- The training time tighter ELBO benefits the performance in terms of avoiding posterior collapse. As we state and empirically verify in Sec. 5.3. We adopt deterministic rounding during inference time, and there is no direct connection between the training time tighter ELBO and inference time R-D cost. However, we indeed end up with a richer latent space (Sec. 5.3), which means more active latent dimensions and less bitrate waste.

- The inference time tighter ELBO sounds really alluring for compression community. However, there remains two pending issue to be resolved prior to the application of the inference time tighter ELBO: 1) How this inference time multiple-sample ELBO is related to R-D cost remains under-explored. In other words, whether the entropy coding itself can achieve the R-D cost defined by multiple-sample ELBO is a question. 2) The inference time multiple-sample ELBO only makes sense with stochastic encoder (you can not importance weight the same deterministic ELBO), whose impact on lossy compression remains dubious.

For the first pending issue, the softmin coding [Theis and Ho, 2021] is proposed to achieve multiple sample ELBO based on Universal Quantization (UQ) [Agustsson and Theis, 2020]. However, it is not a general method and is tied to UQ. Moreover, as we stated in Sec 4.3, its computational cost is forbiddingly high and its improvement is marginal. But those are not the real problem of softmin coding. Instead, the real problem is the second pending issue: stochastic lossy encoder. The softmin coding relies on UQ, and UQ relies on stochastic lossy encoder. And the stochastic lossy encoder is exactly the second issue that we want to discuss.

### A.7.2 Stochastic Lossy Encoder and Universal Quantization

It is known to lossless compression community that stochastic lossy encoder benefits compression performance [Ryder et al., 2022] with the aid of bits-back coding [Townsend et al., 2018]. While the bits-back coding is not applicable to lossy compression. For lossy compression, currently we know that the stochastic encoder degrades R-D performance especially when distortion is measured in MSE [Theis and Agustsson, 2021]. In the original UQ paper, the performance decay of vanilla UQ over deterministic rounding is obvious ($\approx$ 1db). When we writing this paper, we also find the performance decay of UQ is quite high. As shown in Tab. 11, the R-D cost of UQ is significantly higher than deterministic rounding. This negative result makes softmin coding less promising than it seems as it only obtains a marginal gain over UQ. In our humble opinion, this performance decay

|  | y bpp | z bpp | MSE | RD Cost |
|---|---|---|---|---|
| Deterministic Rounding | 0.3347 | 0.01418 | 26.86 | 0.7552 |
| Universal Quantization | 0.5379 | 0.01431 | 23.94 | 0.9080 |

Table 11: The R-D performance of UQ vs deterministic rounding on the first image of Kodak dataset.

of UQ is partially brought by stochastic encoder itself. For lossless compression, the deterministic encoder and stochastic encoder are just two types of bit allocation preference:

- The deterministic encoder allocate less bitrate to $\log p(\tilde{\boldsymbol{y}})$, more to $\log p(\boldsymbol{x}|\tilde{\boldsymbol{y}})$ and $0$ to $\log q(\tilde{\boldsymbol{y}}|\boldsymbol{x})$.

- Given the same quantization step-size, from the differential entropy's perspective, the stochastic encoder allocate more bitrate to $\log p(\tilde{\boldsymbol{y}})$, less to $\log p(\boldsymbol{x}|\tilde{\boldsymbol{y}})$ and minus bitrate to $\log q(\tilde{\boldsymbol{y}}|\boldsymbol{x})$

Therefore, for lossless compression, it is reasonable that the bitrate increase to $\log p(\tilde{\boldsymbol{y}})$ and $\log p(\boldsymbol{x}|\tilde{\boldsymbol{y}})$ can be offset by bits-back coding bitrate $\log q(\tilde{\boldsymbol{y}}|\boldsymbol{x})$. While for lossy compression, there is no obvious way to bits-back $\log q(\tilde{\boldsymbol{y}}|\boldsymbol{x})$ (as we can not reconstruct $q(\tilde{\boldsymbol{y}}|\boldsymbol{x})$ without $\boldsymbol{x}$). If the bitrate increase in $\log p(\tilde{\boldsymbol{y}})$ and $\log p(\boldsymbol{x}|\tilde{\boldsymbol{y}})$, the R-D cost just increases for lossy compression. Prior to other entropy coding bitrate that is able to achieve R-D cost equals to minus ELBO with $\mathbb{E}_q[\log q] \neq 0$ becomes mature (such as relative entropy coding [Flamich et al., 2020]), we have no way to implement a stochastic lossy encoder with reasonable R-D performance. By now, we have no good way to achieve tighter ELBO during inference time.

### A.7.3 Training-Testing Distribution Mismatch and Universal Quantization

Moreover, whether the quantization error is uniform distribution remains a question. And we think that is another reason why UQ does not work well. In fact, the real distribution of quantization noise is pretty much a highly concentrated distribution around $0$ (See Fig. A.7.3). And it is quite far away from uniform distribution, which violates the assumption of Ballé et al. [2018]. We also find that this concentrated distribution is caused by that most of latent dimension is quite close to $0$. The evidence is, if we remove the latent dimension $y^i \in [-0.5, 0.5]$, then the quantization noise looks similar to a uniform distribution (See Fig. A.7.3). So, if we apply direct rounding, they are kept as $0$ and the latent is sparse. However, adding uniform noise to it loses this sparsity, which result in bitrate increase. And from Tab. 11, we can wee that the UQ reduce MSE while increase the bitrate. From total R-D cost perspective, the deterministic rounding outperforms UQ. As a matter of fact, the assumption of UQ that resolving training-testing distribution mismatch improves R-D performance does not hold well. To wrap up, we find that there is some pending issues to be resolved prior to the practical solution of tighter ELBO for inference time.

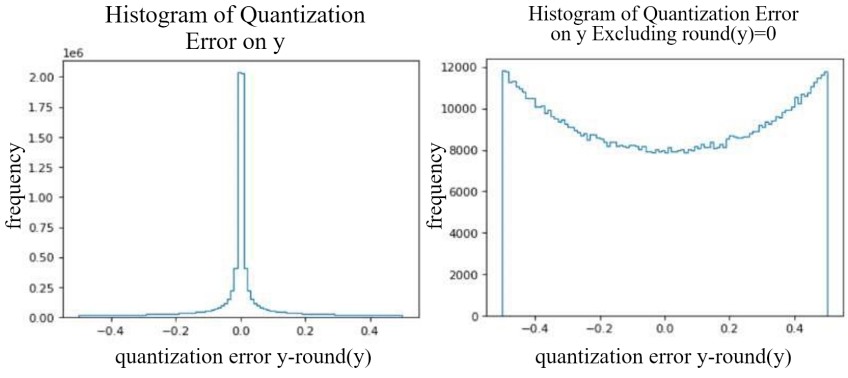

Figure 4: The histogram of $\boldsymbol{y} - \bar{\boldsymbol{y}}$ of first image of Kodak dataset.

## A.8  The Effect on Training Time

The MS-NIC-MIX and MS-NIC-DMS should be more time efficient than simply increase batchsize. For MS-NIC-MIX with $k$ samples, the $\tilde{\boldsymbol{y}}$ encoder $q(\tilde{\boldsymbol{y}}|\boldsymbol{x})$ is inferred with only 1 sample, and the $\tilde{\boldsymbol{z}}$ encoder, decoder and entropy model $q(\tilde{\boldsymbol{z}}|\boldsymbol{x}), p(\tilde{\boldsymbol{y}}|\tilde{\boldsymbol{z}}), p(\tilde{\boldsymbol{z}})$ is inferred with only 1 sample. And only the $\tilde{\boldsymbol{y}}$ decoder $p(\boldsymbol{x}|\tilde{\boldsymbol{y}})$ is inferred $k$ times. This sample efficiency makes the training time grows slowly with $k$. In our experiment, the MS-NIC-MIX with 8 samples only increases the training time by $\times 1.5$, the MS-NIC-MIX with 16 samples only increases the training time by $\times 3$. The MS-NIC-DMS is slightly slower, as the $z$ entropy model and decoder $p(\tilde{\boldsymbol{z}}), p(\tilde{\boldsymbol{y}}|\tilde{\boldsymbol{z}})$ also requires $k$ times inference. However, it is still much more efficient than batchsize $\times k$ as all the encoders $q(\tilde{\boldsymbol{y}}|\boldsymbol{x}), q(\tilde{\boldsymbol{z}}|\boldsymbol{x})$ requires only 1 inference. In fact, sampling from posterior is much cheaper than inferring the posterior parameters. Similar spirit has also been adopted in improving the efficiency of sampling from Gumbel-Softmax relaxed posterior [Paulus et al., 2020].

The trade-off between batchsize and sample number is a more subtle issue. As stated in Rainforth et al. [2018], the gradient SNR of encoder (inference model) scales with $\Theta(M/K)$, and the gradient SNR of decoder (generative model) scales with $\Theta(M/K)$, where $M$ is the batchsize and $K$ is the sample size. Another assumption required prior to further discussion is that the suboptimality of VAE mainly comes from inference model [Cremer et al., 2018], which means that the encoder is harder to train than the decoder. This means that an infinitely large $K$ ruins the convergence of encoder, and solemnly increasing sample number frustrates training. In practice the overall performance is determined by both inference suboptimality and ELBO-likelihood gap. In a word, we believe there is no general answer for all problem. But a reasonable balance between sample size and batchsize is the golden rule to maximize performance (as $T(M)$ and $T(K)$ grow linearly with batchsize/sample size). And the obvious case is that neither setting batchsize to $M = 1$ and give all resources to $K$, nor setting sample size $K = 1$ and give all resources to $M$ is optimal.

## A.9  More Limitation and Discussion

The cause of negative results on MS-SSIM of Cheng et al. [2020] is more complicated. One possible explanation is that the gradient property of Cheng et al. [2020] is not as good as Ballé et al. [2018]. As a reference, the training of [Burda et al., 2016] totally fails on Cheng et al. [2020] and produces garbage R-D results (See Tab. 9). This bad gradient property might account for the bad results of MS-SSIM on Cheng et al. [2020], as the gradient of IWAE and MS-NIC is certainly trickier than the gradient of single sample approaches.

As evidence, when we are studying the stability of the network in Cheng et al. [2020], we find that without limitation of entropy model (imagine setting $\lambda$ to $\infty$) and quantization, Cheng et al. [2020] produces PSNR of $43.27$db, while Ballé et al. [2018] produces PSNR of $48.54$db. This means that Cheng et al. [2020] is not as good as Ballé et al. [2018] as an auto-encoder. Moreover, when we finetune these pre-trained model into a lossy compression model, Cheng et al. [2020] produces $nan$ results while Ballé et al. [2018] converges. This result indicates that the backbone of Cheng et al. [2020]'s gradient is probably more difficult to deal with than Ballé et al. [2018].

## A.10  Broader Impact

Improving the R-D performance of NIC methods is valuable itself. It is beneficial to reducing the carbon emission by reducing the resources required to transfer and store images. And NIC has potential of saving network channel bandwidth and disk storage over traditional codecs. Moreover, for traditional codecs, usually dedicated hardware accelerators are required for efficient decoding. This codec-hardware bondage hinders the wide adaptation of new codecs. Despite the sub-optimal R-D performance of old codecs such as JPEG, H264, they are still prevalent due to broad hardware support. While modern codecs such as H266 [Bross et al., 2021] can not be widely adopted due to limited hardware decoder deployment. However, for NIC, the general purpose neural processors are able to fit to all codecs. Thus the neural decoders have better hardware flexibility, and the cost to update neural decoder only involves software, which encourages the adoption of newer methods with better R-D performance.