# OpenReview forum: "Multi-Sample Training for Neural Image Compression"
_NeurIPS.cc/2022/Conference — NeurIPS 2022 Accept_

### Official Review · Reviewer_5bm9 · 2022-07-11

**Rating:** 6
**Confidence:** 2
**Soundness:** 3 good
**Presentation:** 4 excellent
**Contribution:** 3 good

**Summary:**

Neural Image Compression (NIC) is typically formulated with a VAE objective for optimization, with the rate-distortion objective corresponding to the ELBO, but with some pecularities: the posterior of the inference network is assumed to be uniformly distributed to simulate quantization noise. With this in mind, the paper carefully studies a) the impact of this on various gradient estimators and b) how to properly adapt multi-sample objectives such as the IWAE target.
The proposed MS-NIC objective results in 2-5% BD-rate savings in terms of PSNR for two well known approaches (Balle et al, 2018) and (Cheng et al, 2020).

**Questions:**

See above.

**Limitations:**

Yes.

**Strengths And Weaknesses:**

Note: While very familiar with neural image compression, I am not an expert on multi-sample VAEs and gradient estimators.

I found the paper well written. It gives a very detailed analysis on how the NIC objective connects to VAEs, going into the details such as the direct-y trick of the hyperprior, how the unform posterior is discontinuous and how that affects various gradient estimators.

The resulting MS-NIC objective seems well motivated to me and the empirical results are promising, given that only the training objective has been changed. This means we get better systems with the exactly same architecture.

However, this also brings me to point of confusion: as far as I understand, the multi sample objective is training-only, while at inference time we use the standard single-forward-pass encoder and quantization.
So do we have any guarantees that the "tighter ELBO" that we get from the multi-sample NIC objective  actually improves the inference objective?

The other concern is how the quantization vs noise mismatch interacts with the whole analysis, as indeed the NIC objective is only a proxy for quantization (while it can be implemented at test time with Universal Quantization).

A third concern is how the proposed objective affects training time. Because another way to improve results is to train longer or with a larger batch size, so it would be interesting to understand the trade-offs there.

---

> ### Author Response · Authors · 2022-07-31
> **Response to the comments from Reviewer 5bm9 Part I**
>
> Thanks for your detailed review. And we are glad to provide our answer to your questions, and a few clarification on some misunderstandings:
>
> ### Q1 do we have any guarantees that the "tighter ELBO" that we get from the multi-sample NIC objective actually improves the inference objective?
> * This is a very interesting question. The inference time tighter ELBO is another under-explored issue. In fact, the training time tighter ELBO and inference time ELBO is independent. We can train a model with tighter ELBO, infer with single sample ELBO. Or we can also conduct multiple sample infer on a model trained with single sample. The general idea is:
>   * The training time tighter ELBO benefits the performance in terms of avoiding posterior collapse. As we state and empirically verify in Sec. 5.3. We adopt deterministic rounding during inference time, and there is no direct connection between the training time tighter ELBO and inference time R-D cost. However, we indeed end up with a richer latent space (Sec. 5.3), which means more active latent dimensions and less bitrate waste.
>   * The inference time tighter ELBO sounds really alluring for compression community. However, there remains two pending issue to be resolved prior to the application of the inference time tighter ELBO:
>     * How this inference time multiple-sample ELBO is related to R-D cost remains under-explored. In other words, whether the entropy coding itself can achieve the R-D cost defined by multiple-sample ELBO is a question.
>     * The inference time multiple-sample ELBO only makes sense with stochastic encoder (you can not importance weight the same deterministic ELBO), whose impact on lossy compression remains dubious.
> * For the first pending issue, the softmin coding [Theis and Ho 2021] is proposed to achieve multiple sample ELBO based on Universal Quantization (UQ) [Agustsson and Theis, 2020]. However, it is not a general method and is tied to UQ. Moreover, as we stated in Sec 4.3, its computational cost is forbiddingly high and its improvement is marginal. But those are not the real problem of softmin coding. Instead, the real problem is the second pending issue: stochastic lossy encoder. The softmin coding relies on UQ, and UQ relies on stochastic lossy encoder.
> * It is known to lossless compression community that stochastic lossy encoder benefits compression performance [Theis and Agustsson, 2021] with the aid of bits-back coding [Townsend et al., 2018]. While the bits-back coding is not applicable to lossy compression. For lossy compression, currently we know that the stochastic encoder degrades R-D performance especially when distortion is measured in MSE [Ryder et al., 2022]. In the original UQ paper, the performance decay of vanilla UQ over deterministic rounding is obvious (~1 db). When we writing this paper, we also find the performance decay of UQ is quite high:
>
>   |                        | y bpp  | z bpp   | MSE   | RD Cost |
>   | ---------------------- | ------ | ------- | ----- | ------- |
>   | Deterministic Rounding | 0.3347 | 0.01418 | 26.86 | 0.7552  |
>   | Universal Quantization | 0.5379 | 0.01431 | 23.94 | 0.9080  |
>
> * We can see that compared with deterministic rounding, UQ's bpp ($\log p(y)$ in ELBO) is higher, and MSE ($\log p(x|y)$ in ELBO) is lower. In general, we find that UQ's performance decay is much more significant than the potential performance gain by softmin coding.
> * In our humble opinion, this performance decay is brought by stochastic encoder itself. For lossless compression, the deterministic encoder and stochastic encoder are just two types of bit allocation preference:
>   * The deterministic encoder allocate less bitrate to $\log p(y)$, more to $\log p(x|y)$ and $0$ to $\log q(y|x)$.
>   * The stochastic encoder allocate more bitrate to $\log p(y)$, less to $\log p(x|y)$ and minus bitrate to $\log q(y|x)$
> * Therefore, for lossless compression, it is reasonable that the bitrate increase to $\log p(y)$ and $\log p(x|y)$ can be offset by bits-back coding bitrate $\log q(y|x)$. While for lossy compression, there is no way to bits-back $\log q(y|x)$ (as we can not reconstruct $q(y|x)$ without $x$). If the bitrate increase in $\log p(y)$ and $\log p(x|y)$, the R-D cost just increases for lossy compression. Prior to other entropy coding bitrate that is able to achieve R-D cost equals to minus ELBO with $E_q[\log q]!=0$ becomes mature (such as relative entropy coding [Flamich et al., 2020]), we have no way to implement a stochastic lossy encoder with reasonable R-D performance. By now, we have no good way to achieve tighter ELBO during inference time.
> * __For revision__: we will include those discussions.

---

> ### Author Response · Authors · 2022-07-31
> **Response to the comments from Reviewer 5bm9 Part II**
>
> ### Q2 how the quantization vs noise mismatch interacts with the whole analysis
> * That is another really interesting problem. As we stated above, the benefits of our method comes from avoiding posterior collapse instead of achieving tighter ELBO during inference time. We are quite aware of the issue of the training/inference GAP of $q(y|x)$. And that is the reason why we provide analysis of latent space in both continuous (AUN) case and discrete (quantization) case (See Sec. 5.3). It is shown that the multiple sample latent is consistently richer in both continuous space and discrete space, which means that our method is effective regardless the mismatch.
> * However, as far as we know, this training/inference mismatch is more severe than the UQ paper has described. We examine the quantization error of $y$, and we are surprised to find that the quantization error distribution is not a uniform distribution. Instead, its histogram has a slab-and-spike shape (See histogram in: https://ibb.co/GxkwvNT). This histogram indicates that the basic assumption that quantization error is a uniform distribution is dubious. We believe this mismatch is another source why UQ does not work well when distortion is measured in MSE.
> * __For revision__: we will include those discussions.

---

> ### Author Response · Authors · 2022-07-31
> **Response to the comments from Reviewer 5bm9 Part III**
>
> ### Q3 How the proposed objective affects training time
> * The MS-NIC-MIX and MS-NIC-DMS is more time efficient (in terms of FLOPS/MACS) than simply increase batchsize. For MS-NIC-MIX with $k$ samples, the $y$ encoder $q(y|x)$ is inferred with only 1 sample, and the $z$ encoder, decoder and entropy model $q(z|x),p(y|z),p(z)$ is inferred with only 1 sample. And only the $y$ decoder $p(x|y)$ is inferred $k$ times. This sample efficiency makes the training time grows slowly with $k$. In our experiment, the MS-NIC-MIX with 8 samples only increases the training time by $\times 1.5$, the MS-NIC-MIX with $16$ samples only increases the training time by $\times 3$. The MS-NIC-DMS is slightly slower, as the $z$ entropy model and decoder $p(z),p(y|z)$ also requires $k$ times inference. However, it is still much more efficient than batchsize $\times k$ as all the encoders $q(y|x),q(z|x)$ requires only 1 inference. In fact, sampling from posterior is much cheaper than inferring the posterior parameters. Similar spirit has also been adopted in improving the efficiency of sampling from Gumbel-Softmax relaxed posterior [Paulus et al., 2020].
> * From the above discussions, one might conclude that we should use as much multiple samples as possible and set batchsize to $1$. However, further investigation is required. However, the trade-off between batchsize and sample number might be much more subtle and complicated despite its innocent look. As stated in [Rainforth et al., 2018], the gradient SNR of encoder (inference model) scales with $\Theta(MK)$, and the gradient SNR of decoder (generative model) scales with $\Theta(M/K)$, where M is the batchsize and K is the sample size. Another assumption required prior to further discussion is that the suboptimality of VAE mainly comes from inference model [Cremer et al. 2018], which means that the encoder is harder to train than the decoder. With those assumption in mind, combined with the sample efficiency between our approach and batchsize, we can formulate the problem of finding optimal batshsize $M$, sample size $K$, given limited time $T$, into a quadratic integer programming problem: $M,K\leftarrow\arg\max MK, s.t. T(M)+T(K)<T$, where $T(M)$ and $T(K)$ are the time spend on $M$ batchsize and $K$ multiple sample size, and $T$ is the overall time limit. That is the simplistic case when we exclude the effect of tighter ELBO and consider inference suboptimality only. In practice the overall performance is determined by both inference suboptimality and ELBO-likelihood gap. In a word, we believe there is no general answer for all problem. But a reasonable balance between sample size and batchsize is the golden rule to maximize performance (as $T(M)$ and $T(K)$ grow linearly with batchsize/sample size). And the obvious case is that neither setting batchsize to $M=1$ and give all resources to $K$, nor setting sample size $K=1$ and give all resources to $M$ is optimal.
>
> * __For revision__: we will include those discussions.
>
> Reference:
> * J. Townsend, T. Bird, and D. Barber. Practical lossless compression with latent variables using bits back coding. In International Conference on Learning Representations, 2018
> * L. Theis and E. Agustsson. On the advantages of stochastic encoders. arXiv preprint arXiv:2102.09270, 2021.
> * T. Ryder, C. Zhang, N. Kang, and S. Zhang. Split hierarchical variational compression. In Proceedings of the IEEE/CVF Conference on Computer Vision and Pattern Recognition, pages 386–395, 2022.
> * M. B. Paulus, C. J. Maddison, and A. Krause. Rao-blackwellizing the straight-through gumbel-365
> softmax gradient estimator. arXiv preprint arXiv:2010.04838, 2020.
> * G. Flamich, M. Havasi, and J. M. Hernández-Lobato. Compressing images by encoding their latent representations with relative entropy coding. Advances in Neural Information Processing Systems,33:16131–16141, 2020.
> * T. Rainforth, A. Kosiorek, T. A. Le, C. Maddison, M. Igl, F. Wood, and Y. W. Teh. Tighter variational bounds are not necessarily better. In International Conference on Machine Learning, pages 4277–4285. PMLR, 2018.
> * C. Cremer, X. Li, and D. Duvenaud. Inference suboptimality in variational autoencoders. In International Conference on Machine Learning, pages 1078–1086. PMLR, 2018.

---

> ### Comment · Reviewer_5bm9 · 2022-08-09
> **Response**
>
> I appreciate the thorough response of the authors, which addresses most of my concerns and adds an interesting discussion.
>
> However, I find the additional appendix A.11.3 to unfairly characterise UQ.
>
> The authors write in l.658 of the appendix:
> "
> As a matter of fact, the assumption of UQ that resolving training-testing distribution mismatch improves R-D performance is dubious.
> "
>
> I find this claim really strange, given that in the UQ paper (https://arxiv.org/pdf/2006.09952.pdf, sec 5.3) it is written:
> """
> When comparing the UN + UQ model which uses universal quantization to the test-time quantization baseline UN + Q, we see that despite the train-test mismatch using quantization improves the RD-performance at test-time (hatched area).
> """
>
> So this has already been observed and discussed in the original UQ paper, so what exactly is dubious?
>
> Clearly there is a train-test mismatch as the authors have also observed (plain rounding does not give a uniform noise distribution as used in the noise proxy of Balle et al). The UQ paper shows that while it can be eliminated, performance drops unless soft-rounding is introduced.

---

> > ### Author Response · Authors · 2022-08-10
> > **Reply to Reviewer 5bm9**
> >
> > Hi, thanks for your feedback.
> >
> > We are trying to express that "UN+UQ" performs much worse than "UN+Q", which is aligned with Fig. 2A of [Agustsson and Theis, 2020]. And we mean that "UQ improves R-D performance." is dubious, not the original UQ paper is dubious.
> >
> > I think the word "dubious" is abused here. We will amend this in future version.
> >
> > ### Reference:
> > E. Agustsson and L. Theis. Universally quantized neural compression. Advances in neural information processing systems, 33:12367–12376, 2020.

---

### Official Review · Reviewer_8LQV · 2022-07-11

**Rating:** 7
**Confidence:** 4
**Soundness:** 4 excellent
**Presentation:** 3 good
**Contribution:** 3 good

**Summary:**

Most work on lossy neural image compression (NIC) us a "single-sample pathwise estimator" to estimate the ELBO gradients to optimize the core rate-distortion loss function. This paper observes that multiple-sample methods provide a tighter bound than ELBO and thus may lead to better compression results (a lower rate-distortion loss). The paper explores different formulations (IWAE, MS-NIC-MIX, and MS-NIC-DMS) that incorporate multiple samples, offers an explanation for why IWAE does not work here (\tilde{z} depends on \tilde{y} which increases gradient variance compared to models where \tilde{z} depends on y), and shows empirically that both multi-sample methods (MIX and DMS) boost compression performance in most cases (the exception is an autoregressive entropy model optimized for MS-SSIM).

Glossary:
ELBO = evidence lower bound
IWAE = importance-weighted autoencoder
MS = multi-sample
MIX = combine multi and single-sampling
DMS = double multi-sampling
MS-SSIM = multiscale structural similarity


**Questions:**

Presumably, the training time for MS-NIC-DMS will be higher per step compared to the baseline single-sample approach. If so, does multi-sample optimization make sense with a fixed training budget? In other words, if I have a compute (or time) budget for training, should I prefer fewer MS steps or more SS steps?

**Limitations:**

I don't see any potential negative societal impacts from this work. The authors did discuss limitations (the poor results on MS-SSIM with the (Cheng 2020) model is the primary limitation).

**Strengths And Weaknesses:**

Strengths:
1. This paper looks at a fundamental aspect of neural image compression (tighter bounds than ELBO when uniform noise is used for latents) that, as far as I know, has not be previously discussed. The discussion of multi-sample methods is thorough.

2. The approach is quite general in the sense that it is applicable to most neural image compression methods (models that use VQ are an exception) and the empirical benefits are non-trivial (2.38% - 4.93% rate savings for MS-NIC-DMS when optimizing for PSNR). For reference, a 10% rate gain is considered huge, less than 1% is not very interesting since it's on par with the random variation across training runs, and a new generation of a standard codec typically provides a 30% rate savings).

Weaknesses:
1. It's not obvious to me why multi-sample methods would perform worse on MS-SSIM and the (Cheng 2020). Additional exploration is warranted (as the authors say in the conclusion).

2. Confidence intervals for the empirical results would strengthen the paper. The authors do address this in the "Checklist" section, and I agree that training models can be expensive and "error bars" are not commonly seen in NIC papers (though they should be). Perhaps a reasonable compromise would be to estimate the confdience interval in just a single case (e.g. PSNR for (Balle 2018).

3. Either actual code or more guidance on how to implement MS-NIC-DMS would strengthen the paper. I think I can figure this out by reviewing the IWAE and SGVB algorithms.

---

> ### Author Response · Authors · 2022-07-31
> **Response to the comments from Reviewer 8LQV Part I**
>
> ### W1 Why multi-sample methods would perform worse on MS-SSIM and the [Cheng et al. 2020]?
> * One possible explanation is that the gradient property of [Cheng et al. 2020] is not as good as [Ballé et al. 2018]. As a reference, the training of [Burda et al., 2016] totally fails on [Cheng et al. 2020] and produces garbage R-D results (See Tab. 10). This bad gradient property might account for the bad results of MS-SSIM on [Cheng et al. 2020], as the gradient of IWAE and MS-NIC is certainly tricker than the gradient of single sample approaches.
> * As evidence, when we are studying the stability of the network in [Cheng et al. 2020], we find that without limitation of entropy model (imagine setting $\lambda$ to $\infty$) and quantization, [Cheng et al. 2020] produces $nan$ before $2000$ epochs of training while [Ballé et al. 2018] converges properly. Under such setting, [Ballé et al. 2018] achieve PSNR of $48.54$db, while the best of [Cheng et al. 2020] is $43.27$db. This result indicates that the backbone of [Cheng et al. 2020] is not as good as [Ballé et al. 2018] as an auto-encoder. Moreover, when we finetune these pre-trained model into a lossy compression model, Cheng et al. [2020] produces nan results while Ballé et al. [2018] converges. This result indicates that the backbone of Cheng et al. [2020]’s gradient is probably more difficult to deal with than Ballé et al. [2018].
> * __For revision__: we will include those discussions.
> ### W3 Either actual code or more guidance on how to implement MS-NIC-DMS would strengthen the paper. I think I can figure this out by reviewing the IWAE and SGVB algorithms.
> * We are glad to provide the detailed pytorch-style sudo code for implementation guidance. We extract and rewrite the core code for computing the loss of MS-NIC-DMS/MS-NIC-MIX.
> ```python
> import torch
> from torch.nn import functional as F
>
> def IWAELoss(minus_elbo):
>     '''
>     args
>     ----
>     minus_elbo: tensor, [b, k], which is R + \lambda D
>
>     return
>     ------
>     local iwae loss
>     '''
>     # this is the minus ELBO related to y part, to get the real ELBO:
>     log_weights = - minus_elbo.detach() # no gradient given to weights
>     weights = F.softmax(log_weights, dim=1) # B, K
>     loss_b = torch.sum(minus_elbo * weights, dim=1, keepdim=False) # B
>     loss_iwae = torch.mean(loss_b)
>     return loss_iwae
>
> def DMSLoss(x, x_hat, y_likelihood, z_likelihood, lam):
>     '''
>     args
>     ----
>     x: original image: [b, c, h, w]
>     x_hat: reconstructed image: [b, k, c, h, w], k is the number of samples
>     y_likelihood: [b, 192/320, h//8, w//8, k^2], as original paper of [Ballé et al. 2018], the number of channels 192/320 is determined by lambda, k^2 is the number of samples in DMS setting, with MS-NIC-MIX, this k^2 is k
>     z_likelihood: [b, 128/192, h//64, w//64, k], as original paper of [Ballé et al. 2018], the number of channels 128/192 is determined by lambda, k is the number of samples
>
>     return
>     ------
>     total iwae loss
>     '''
>     b, c, h, w = x.shape
>     k = x_hat.shape[0] // x.shape[0]
>     x = torch.repeat_interleave(x, repeats=k, dim=0)
>     x = x.reshape(b, k, c, h, w)
>     x_hat = x_hat.reshape(b, k, c, h, w)
>     d_loss = torch.mean(lam * 65025 * (x - x_hat)**2, dim=(2,3,4), keepdim=False)
>     yz_loss = -torch.sum(torch.log2(y_likelihood), dim=(1,2,3)).reshape(b, -1) / (h * w)
>     z_loss = -torch.sum(torch.log2(z_likelihood), dim=(1,2,3)).reshape(b, -1) / (h * w)
>     local_d = IWAELoss(d_loss)
>     local_yz = IWAELoss(yz_loss)
>     local_z = IWAELoss(z_loss)
>     loss_total = local_d + local_yz + local_z
>
>     return loss_total
> ```
> * __For revision__: we will include those discussions.

---

> ### Author Response · Authors · 2022-07-31
> **Response to the comments from Reviewer 8LQV Part II**
>
> ### Q1 should I prefer fewer MS steps or more SS steps ?
> * This is a quite interesting question, yet it is quite subtle and complicated. First, let's assume that M single sample, single batchsize step is inferior to 1 single sample, M batchsize step [Schmidt, 2019]. The intuition is that 1) ignoring the effect of batchsize to gradient variance, M single batchsize step is equivalent to 1 M batchsize step. 2) the error of gradient descent grows linearly to the variance of gradient estimation, and the variance of gradient estimation grows inversely with batchsize M.
> *  Then the question becomes, given a time budget T, how to allocate the batchsize M and the sample size K. The general rule is, the optimal allocation lies where M and K are not too far away. (In our experiment, we set M=8, and K=1,3,8,16). Perhaps surprisingly, setting either M=1 or K=1 is suboptimal. The exact optimal allocation can be really complicated and effected by multiple factors. (This discussion is similar to the discussion with Reviewer 5bm9 Q3.)
> * The first argument to make is that, the MS-NIC-MIX and MS-NIC-DMS is more time efficient than simply increasing the batchsize (in terms of FLPOS/MACS). For MS-NIC-MIX with $K$ samples, the $y$ encoder $q(y|x)$ is inferred with only 1 sample, and the $z$ encoder, decoder and entropy model $q(z|x),p(y|z),p(z)$ is inferred with only 1 sample. And only the $y$ decoder $p(x|y)$ is inferred $K$ times. This sample efficiency makes the training time grows slowly with $k$. In our experiment, the MS-NIC-MIX with 8 samples only increases the training time by $\times 1.5$, the MS-NIC-MIX with $16$ samples only increases the training time by $\times 3$. The MS-NIC-DMS is slightly slower, as the $z$ entropy model and decoder $p(z),p(y|z)$ also requires $k$ times inference. However, it is still much more efficient than batchsize $\times k$ as all the encoders $q(y|x),q(z|x)$ requires only 1 inference. In fact, sampling from posterior is much cheaper than inferring the posterior parameters. Similar spirit has also been adopted in improving the efficiency of sampling from Gumbel-Softmax relaxed posterior [Paulus et al., 2020].
> * From the above discussions, one might conclude that we should use as much multiple samples as possible and set batchsize to $1$. However, further investigation is required. As stated in [Rainforth et al., 2018], the gradient SNR of encoder (inference model) scales with $\Theta(MK)$, and the gradient SNR of decoder (generative model) scales with $\Theta(M/K)$, where M is the batchsize and K is the sample size. Another assumption required prior to further discussion is that the suboptimality of VAE mainly comes from inference model [Cremer et al. 2018], which means that the encoder is harder to train than the decoder. With those assumption in mind, combined with the sample efficiency between our approach and batchsize, we can formulate the problem of finding optimal batshsize $M$, sample size $K$, given limited time $T$, into a quadratic integer programming problem: $M,K\leftarrow\arg\max MK, s.t. T(M)+T(K)<T$, where $T(M)$ and $T(K)$ are the time spend on $M$ batchsize and $K$ multiple sample size, and $T$ is the overall time limit. That is the simplistic case when we exclude the effect of tighter ELBO and consider inference suboptimality only. In practice the overall performance is determined by both inference suboptimality and ELBO-likelihood gap. In a word, we believe there is no general answer for all problem. But a reasonable balance between sample size and batchsize is the golden rule to maximize performance (as $T(M)$ and $T(K)$ grow linearly with batchsize/sample size). And the obvious case is that neither setting batchsize to $M=1$ and give all resources to $K$, nor setting sample size $K=1$ and give all resources to $M$ is optimal.
> * __For revision__: we will include those discussions.
>
> ### Reference
> * J. Ballé, D. Minnen, S. Singh, S. J. Hwang, and N. Johnston. Variational image compression with a scale hyperprior. In International Conference on Learning Representations, 2018.
> * Z. Cheng, H. Sun, M. Takeuchi, and J. Katto. Learned image compression with discretized gaussian mixture likelihoods and attention modules. In Proceedings of the IEEE/CVF Conference on Computer Vision and Pattern Recognition, pages 7939–7948, 2020.
> * Mark Schmidt, Lecture Notes on CPSC 540 - Machine Learning https://www.cs.ubc.ca/~schmidtm/Courses/540-W19/ 2019
> * M. B. Paulus, C. J. Maddison, and A. Krause. Rao-blackwellizing the straight-through gumbel-365
> * T. Rainforth, A. Kosiorek, T. A. Le, C. Maddison, M. Igl, F. Wood, and Y. W. Teh. Tighter variational bounds are not necessarily better. In International Conference on Machine Learning, pages 4277–4285. PMLR, 2018.
> * C. Cremer, X. Li, and D. Duvenaud. Inference suboptimality in variational autoencoders. In International Conference on Machine Learning, pages 1078–1086. PMLR, 2018.

---

> > ### Comment · Reviewer_8LQV · 2022-08-09
> > **thorough comments that support the 7 ("accept") rating**
> >
> > I appreciate that the authors took the time to write a thorough response (here and for the other reviewers). The comments addressed my questions/concerns, and the added pseudocode also adds clarity to the paper. I maintain my rating of 7 and support accepting this paper.

---

### Official Review · Reviewer_XrNr · 2022-07-11

**Rating:** 6
**Confidence:** 4
**Soundness:** 3 good
**Presentation:** 3 good
**Contribution:** 3 good

**Summary:**

This paper proposes to train neural compression VAEs with multi-sample IWAE-style objectives, and analyzes the effect of the choice of variational distribution (constant width uniform posterior; the choice between q(y|x) v.s. q(y|z)) on various gradient estimators. They theoretically showed that the uniform posterior in neural compression automatically implements a form of the STL estimator [Roeder et al., 2017], and that naively applying the score function gradient estimator would give incorrect results. They propose two IWAE-style objectives that are potentially tighter than the standard NELBO objective, and show empirically that training on them yield small improvement in R-D performance (5% rate savings).

**Questions:**

There are a few claims in this paper that I'm not unsure about.

1. I'm not following the discussions around line 153, regarding why the direct-y trick poses an issue for IWAE. Here we simply have a factorized posterior q(z, y|x) = q(z|x) q(y|x), so why can't we just use k samples of (z, y) from this joint posterior to form an IWAE estimator?

2. The authors' statements about IWAE estimator converging to log p(x) (e.g., line 187-189) can be misleading, as they may not hold for the neural compression VAEs. Again this has to do with the posterior having bounded support, whereas the prior and likelihood models having full support.

3. Is there an obvious explanation for why the mult-sample IWAE [Burda et al., 2016] baseline is so much worse in Table 5? I could not find it.

3. This is a nitpick: the statement  "q(z˜|y˜) and q(y˜|x˜) are not absolute continuous" is technically incorrect (line 103), although I understand the authors meant to say they don't satisfy the "absolute continuity condition of [Mohamed 2020]". The uniform posterior distributions *are* absolutely continuous (w.r.t. Lebesgue measure) since they have densities.  To be fair, the language used in [Mohamed 2020] is also confusing in this regard.

**Limitations:**

It would be helpful to also discuss the computation/implementation complexity of the multi-sample approach.

**Strengths And Weaknesses:**

Strengths:

The paper is mostly clear and easy to follow; the proposed method is somewhat original (see Theis and Ho [2021]) and conceptually straightforward. Even though the experimental results are somewhat negative, I believe a better understanding between variational inference and compression would be of significant interest  to researchers in both fields.


Weaknesses:

1. The authors made several observations regarding different gradient estimators for IWAE, specialized to the VI setup of neural image compression (NIC), but it's unclear what the impact/significance they have.  The first observation (sec 2.1; "STL is unbiased because in NIC the posterior entropy is constant") seems trivial, and the second observation (sec 2.2; score function gradient doesn't work because the NIC posterior has bounded support) also seems straightforward after checking the conditions for when the score function gradient apply ([Mohamed 2020]). As for impact, the first observation appears to be a sanity check for the proposed multi-sample bound, and the second observation appears to caution against an approach (score function gradient estimator) which nobody seems to be using in neural compression ...

2. The experimental results are not very informative/satisfying.  It's perhaps unsurprising that training a NIC model with a tighter bound does not yield significantly improved compression performance, unless an accompanying encoding/inference scheme is used to realize such a bound. It would be interesting to see if softmin coding (Theis and Ho [2021]) can fulfill the potential of the proposed multi-sample training scheme, even on a small scale. Similarly, it'd be helpful to provide some insight into why increasing the number of IWAE samples has a limited impact on performance.

3. Also see below about potentially questionable claims.

---

> ### Author Response · Authors · 2022-07-28
> **Response to the comments Reviewer XrNr Part I**
>
> Thanks for your detailed review. And we are glad to provide our answer to your questions, and a few clarification on some misunderstandings:
>
> ### Weakness 1:
>
> * The major result and conclusion of Sec 2.1-2.2 is not that the REINFORCE [Williams, 1992] should not be used (which is obvious), but is that the DReG [Tucker et al., 2018] estimator should not be used (which is much less obvious). We agree that no one would adopt REINFORCE when reparameterization trick is available. However, the DReG estimator at the first glance indeed looks an innocent improvement over IWAE [Burda et al., 2016]. It is a path-wise estimator instead of a score-function estimator. However, its unbiasedness is built upon the equivalence between REINFORCE and reparameterization trick. In fact, we write Sec 2.2 and Sec 2.1 because we first find the weird result that DReG fails (Tab 1). Then, we carefully examine the derivation of DReG and find it is not compatible with uniform posterior with the help of [Mohamed et al., 2020]. Then, we determine to fully discuss this issue and write Sec 2.1 Sec 2.2 to go over the applicability of common gradient estimators to neural compression. We think that Sec 2.1 and 2.2 is informative to neural image compression, not because that the straightforward REINFORCE might be adopted, but because that more advanced estimators such as VIMCO [Mnih and Rezende, 2016] and DReG could be adopted.
> * To be specific, let $x$ be the observed evident and $y$ be the latent, and denote $w_i=\frac{p_{\theta}(x,y_i)}{q_{\phi}(y_i|x)}$. The DReG estimator observes that with reparameterization $y_i=y(\epsilon_i,\phi)$, the total derivative of a $k$ sample IWAE target can be expanded as $\nabla_{\phi} E_{q_{\phi}(y|x)}[\log \frac{1}{k}\sum_{i=1}^{k}w_i]= E_{p(\epsilon)}[\sum_{i=1}^{k}\frac{w_i}{\sum_{j=1}^{k}w_j}(-\frac{\partial}{\partial \phi}\log q_{\phi}(y_i|x)+\frac{\partial \log w_i}{\partial y_i}\frac{\partial y_i}{\partial \phi})]$. The DReG estimator observes that the $\frac{\partial}{\partial \phi}\log q_{\phi}(y_i|x)$ part of this path-wise gradient
> resembles a score function estimator, and propose to further reparameterize it as $E_{q(y_i|x)}[\frac{w_i}{\sum_{j=1}^{k}w_j}\frac{\partial}{\partial \phi}\log q_{\phi}(y_i|x)]= E_{p(\epsilon_i)}[\frac{\partial}{\partial y_i}(\frac{w_i}{\sum_{j=1}^{k}w_j})\frac{\partial y_i}{\partial \phi_i}]$. However, this step requires the equivalence between score function gradient and path-wise gradient: $E_{q_{\phi}(y|x)}[f(y)\frac{\partial}{\partial \phi}\log q_{\phi}(z|x)]= E_{p(\epsilon)}[\frac{\partial f(y)}{\partial y}\frac{\partial y}{\partial \phi}]$. And such equivalence does not hold in general for our case.
>
> * __For revision__: we will add the detailed derivation about why DReG fails in the revised version of paper.

---

> ### Author Response · Authors · 2022-07-28
> **Response to the comments Reviewer XrNr Part II**
>
> ### Weakness 2 part 1:
>
> * The major reason why we did not include result of softmin coding [Theis and Ho 2021] is not within the softmin coding itself, but instead the Universal Quantization (UQ) [Agustsson and Theis, 2020] method it relies on. The original softmin paper only compares the R-D performance of softmin+UQ and UQ, but does not include the deterministic scalar quantization. In fact,  the UQ makes the encoder stochastic, which brings RD performance decay compared with deterministic scalar quantization when distortion is measured by MES. As shown in the original paper, UQ performs worse than scalar quantization without a trick named Soft Rounding. However, the Soft Rounding trick change the variational posterior from uniform distribution towards a spike-like distribution with weights concentrated in integer bins, which makes the softmin coding inapplicable. The performance decay of UQ along looks quite significant (~1 db in PSNR), and the performance gain by softmin coding looks only marginal to us.
>
> * The inference time multi-sample coding only makes sense with stochastic encoder. However, in our humble opinion, the stochastic encoder itself is the reason why UQ does not work. While the stochastic encoder outperforms deterministic encoder in lossless compression with bits-back coding [Townsend et al., 2018], it remain dubious in lossy compression as the entropy of stochastic encoder (the $\log q(y|x)$ term in ELBO) can not be bits-back coded. As the bits-back coding is known to be incompatible to lossy compression. Thus, even if the ELBO of stochastic encoder is tighter than the deterministic encoder, without bits-back coding we can not reduce the $\log q(y|x)$ term from the bit stream. In fact, the stochastic encoder for lossy compression and lossy bits-back coding is quite interesting and relatively under-explored. There are indeed other approaches, such as relative entropy coding (REC) that can achieve the similar effect of lossy bits-back coding. However, it requires another dedicated entropy coding technique to achieve tighter ELBO, just like softmin for UQ. We believe that the stochastic lossy encoder is out of the scope of our paper and deserves a separated dedicated paper to discuss. And we think that the multi-sample coding for inference time is not ready prior to the issue of stochastic lossy encoder is addressed.
>
> * In fact, during the paper writing process, we explored into this problem preliminarily. We find that applying UQ during inference time greatly harms the R-D performance compared with deterministic rounding:
>
> |                        | y bpp  | z bpp   | MSE   | RD Cost |
> | ---------------------- | ------ | ------- | ----- | ------- |
> | Deterministic Rounding | 0.3347 | 0.01418 | 26.86 | 0.7552  |
> | Universal Quantization | 0.5379 | 0.01431 | 23.94 | 0.9080  |
>
> * And we find that the major reason that UQ fails is that during the inference time the quantization noise is not a uniform distribution at all! In fact, the real distribution of quantization noise is pretty much a highly concentrated distribution around $0$ (See histogram in https://ibb.co/GxkwvNT) . And it is quite far away from uniform distribution. We also find that this concentrated distribution is caused by that most of latent dimension is quite close to $0$. The evidence is, if we remove the latent dimension $y^i\in [-0.5,0.5]$, then the quantization noise looks similar to a uniform distribution (See histogram in https://ibb.co/j57QB3T). So, if we apply direct rounding, they are kept as $0$ and the latent is sparse. However, adding uniform noise to it loses this sparsity. As a matter of fact, the assumption of UQ does not holds for inference time. To wrap up, those observations on UQ really hinders us to explore softmin coding during our paper writing.
> * __For revision__: we will extend Sec 4.3 to include these discussions.

---

> ### Author Response · Authors · 2022-07-28
> **Response to the comments Reviewer XrNr Part III**
>
> ### Weakness 2 part 2:
> * We think that the performance improvement of our approach is bounded by how severe the posterior collapse is in neural image compression. We measure the variance in latent dimension according to data in Fig. 3. And from that figure it might be observed that the major divergence of IWAE and VAE happens when the variance is very small. And for the area where variance is reasonably large, the gain of IWAE is not that large. This probably indicates that the posterior collapse in neural image compression is only alleviated to a limited extend.
>
> * __For revision__: we will extend Sec 5.3 to include these discussions.
>
> ### Question 1,3:
>
> * The problem of taking $k$ samples from $q(z,y|x)=q(z|x)q(y|x)$ is that the $q(z|x)$ and $q(y|x)$ is independent. Assuming we sample $y_{1:k}$ from $q(y|x)$, we still only have 1 distribution for $q(z|y)$. Again, if we also draw $z_{1:k}$ from $q(z|x)$, it would actually be a waste of sample to pair $z_1$ with $y_1$, …, $z_k$ with $y_k$, as they are essentially independent. Moreover, the number sample that we draw from $q(y|x)$ and $q(z|x)$ does not need to be the same as $z,y$ factorizes given $x$. Assuming we sample $y_{1:m}$, $z_{1:n}$, a smarter way to pair the samples is to pair each $y$ $n\times z$, which means that we can generate $m\times n$ samples with $m+n$ draws. In fact, this leads MS-NIC-DMS described in Sec 3.2.
>
> * Thus, the plain IWAE sampling from $q(z,y|x)=q(z|x)q(y|x)$ is unreasonable. And the natural condition where IWAE fits is another form of factorization: $q(z,y|x)=q(z|y)q(y|x)$. Now assuming we sample $y_{1:k}$ from $q(y|x)$, we have $k$ distribution for $q(z|y_i)$. However, this type of factorization brings severe performance decay due to the gradient variance issue as we show in Sec 2.3, and the plain IWAE is tied to this type of factorization. That is the reason why it’s performance is much worse in Tab. 5.
>
> ### Question 2:
> * Thanks for pointing this out. Probably the proper expression for 187-188 is:
>   * $\mathcal{L}^{DMS}_{k,l}\xrightarrow{as} \log p(x)$, under the assumption that $p(x|y_i)p(y_i|z_j)/q(y_i|x)$ and $p(x|z_j)p(z_j)/q(z_j|x)$ is bounded.
> * And the proof follows the Strong Low of Large Number instead of the Weak Low of Large Number.
> * __For revision__: we will revise the proposition and proof. We are OK to abandon this claim if you still find it problematic. As it does not harm our central claims.
>
> ### Question 4:
> * Thanks for pointing it out.
> * __For revision__: we will revise this as much as we can.
>
> ### Reference:
> * G. Tucker, D. Lawson, S. Gu, and C. J. Maddison. Doubly reparameterized gradient estimators for monte carlo objectives. 2018.
> * R. J. Williams. Simple statistical gradient-following algorithms for connectionist reinforcement learning. Machine learning, 8(3):229–256, 1992.
> * A. Mnih and D. Rezende. Variational inference for monte carlo objectives. In International Conference on Machine Learning, pages 2188–2196. PMLR, 2016
> * L. Theis and J. Ho. Importance weighted compression. In Neural Compression: From Information Theory to Applications–Workshop@ ICLR 2021, 2021.
> * E. Agustsson and L. Theis. Universally quantized neural compression. Advances in neural information processing systems, 33:12367–12376, 2020.
> * Y. Burda, R. B. Grosse, and R. Salakhutdinov. Importance weighted autoencoders. In ICLR (Poster), 2016.
> * J. Townsend, T. Bird, and D. Barber. Practical lossless compression with latent variables using bits back coding. In International Conference on Learning Representations, 2018

---

> ### Comment · Reviewer_XrNr · 2022-08-10
> **Rebuttal acknowledgement**
>
> I thank the authors for the detailed response, which has addressed most of my concerns and provided interesting new insights to the problem.
>
> I think the paper made a good contribution that connects and fills in the blank of existing literature, and I've raised my score.

---

### Author Response · Authors · 2022-08-02
**Summary of Revision**

Thanks for your detailed review. We have uploaded the revised main text and supplementary material, with all the revisions marked in blue. Due to the space limitation, we could not include all the amendment in the main text. Below is a summary of revisions:

### Main Text
* Sec 2.2: We revise the expression about absolute continuity and expand the discussion on the impact of gradient estimators on IWAE-DReG and add detailed formulas explaining why it does not work. (as suggested by XrNr)
* Sec 3.2: We revise the assumption of convergence condition. (as suggested by XrNr)
* Sec 4.2: We add discussion that stochastic lossy encoding remains dubious, and we provide a pointer to in-depth discussion in Appendix. (as suggested by XrNr,5bm9)
* Sec 5.3: We move a table to appendix to make room for extra discussions.

### Appendix
* Appendix A.4: We revise the assumption and proof of convergence condition.  (as suggested by XrNr)
* Appendix A.9 We add additional discussion on why the performance improvement of MS approach is limited (as suggested by XrNr) and why the MS-SSIM on [Cheng et al. 2020] is negative.
* Appendix A.11.1: A new section discussing inference time ELBO and softmin coding (as suggested by XrNr, 5bm9).
* Appendix A.11.2: A new section discussing Stochastic Lossy Encoder and Universal Quantization (as suggested by XrNr, 5bm9)
* Appendix A.11.3: A new section discussing Training-Testing Distribution Mismatch and Universal Quantization (as suggested by XrNr, 5bm9)
* Appendix A.12: A new section discussing the impact of our method on training time, and how to balance the batchsize and sample size (as suggested by 8LQV, 5bm9)
* Appendix A.13: A new section discussing the implementation guidance (as suggested by 8LQV)

### Reference
* Y. Yang, R. Bamler, and S. Mandt. Improving inference for neural image compression. Advances in Neural Information Processing Systems, 33:573–584, 2020.
* M. Song, J. Choi, and B. Han. Variable-rate deep image compression through spatially-adaptive feature transform. In 2021 IEEE/CVF International Conference on Computer Vision, ICCV 2021, pages 2360–2369. IEEE, 2021.

---

### Comment · Area_Chair_79Fx · 2022-08-07
**Discussion period**

Thank you to all the reviewers for the great effort in reviewing the paper and the authors for the responses.

As in the discussion period, I want to ensure that reviewers have read the authors' responses and engage with the authors if needed.

If you haven't done this, could you please take a moment to read through the authors' responses, update the reviews to indicate that you have read the authors' responses, or communicate with the authors if needed? You can also share in private conversations with the reviewing team.

Please continue to share your thoughts. Thank you!

---

### Meta-Review · Area_Chair_79Fx · 2022-08-28

**Recommendation:** Accept
**Confidence:** Less certain

**Metareview:**

This paper studies the problem of neural image compression (NIC). Standard methods for NIC use a "single-sample pathwise estimator" to estimate the ELBO gradients to optimize the rate-distortion loss function. This paper improves the estimation by using multiple samples, leading to better compression results. Experimental results show that multi-sample methods improve compression performance in many cases.

The reviewers' comments are appropriately addressed, and all reviewers appreciate the contribution of this paper on neural image compression, so I recommend accept.

**Award:**

No

---

### Decision · Program_Chairs · 2022-09-14

Accept